# Balancing Positive and Negative Classification Error Rates in Positive-Unlabeled Learning

**Ximing Li[1,2,3], Yuanchao Dai[1,2], Bing Wang[1,2], Changchun Li[1,2],\* Jianfeng Qu[4,5], Renchu Guan[1,2]**

[1]College of Computer Science and Technology, Jilin University, China
[2]Key Laboratory of Symbolic Computation and Knowledge Engineering, Jilin University, China
[3]RIKEN Center for Advanced Intelligence Project
[4]School of Computer Science and Technology, Soochow University, China
[5]Suzhou Key Lab of Multi-modal Data Fusion and Intelligent Healthcare, Suzhou City University, China
`{liximing86, yuanchaodai, changchunli93}@gmail.com`

## Abstract

Positive and Unlabeled (PU) learning is a special case of binary classification with weak supervision, where only positive labeled and unlabeled data are available. Previous studies suggest several specific risk estimators of PU learning such as non-negative PU (nnPU), which are unbiased and consistent with the expected risk of supervised binary classification. In nnPU, the negative-class empirical risk is estimated by positive labeled and unlabeled data with a non-negativity constraint. However, its negative-class empirical risk estimator approaches 0, so the negative class is over-played, resulting in imbalanced error rates between positive and negative classes. To solve this problem, we suppose that the expected risks of the positive-class and negative-class should be close. Accordingly, we constrain that the negative-class empirical risk estimator is lower bounded by the positive-class empirical risk, instead of 0; and also incorporate an explicit equality constraint between them. We suggest a risk estimator of PU learning that balances positive and negative classification error rates, named DC-PU, and suggest an efficient training method for DC-PU based on the augmented Lagrange multiplier framework. We theoretically analyze the estimation error of DC-PU and empirically validate that DC-PU achieves higher accuracy and converges more stable than other risk estimators of PU learning. Additionally, DC-PU also performs competitive accuracy performance with practical PU learning methods.

## 1 Introduction

**P**ositive and **N**egative (**PN**) learning refers to conventional supervised binary classification trained with positive and negative labeled data [1]. **P**ositive and **U**nlabeled (**PU**) learning is a specific case of PN learning with weak supervision [2, 3], where, as its name suggests, it trains a binary classifier with only positive labeled and unlabeled data but negative labeled data are unknown [1]. The paradigm arises in various real-world scenarios, such as outlier detection and information retrieval [4, 5]. During the past decade, there have been many practical PU learning methods mainly estimating pseudo-labels for unlabeled data [6, 7, 8, 9]; and another research branch of PU learning is to formulate specific risk estimators [10, 11, 12, 13], while some of them can be unbiased and consistent with the expected risk of PN learning.

To distinctly discuss the risk estimators, we first give the problem assumption of PU learning, and in this work, we concentrate on the *two-sample problem setting* [14]. To be specific, positive data are

---

\*Corresponding author

drawn separately from unlabeled data. The *completely selected at random* assumption is typically used, *i.e.* the positive labeled data are identically distributed as positive unlabeled data, so the positive labeled data are drawn from the positive-class conditional distribution, and the unlabeled data are drawn from the whole population [11, 12]. Upon this setting, [11] suggests an unbiased risk estimator of PU learning named uPU, which, specifically, contains a positive-class empirical risk with positive labeled data and an unbiased negative-class empirical risk estimated by positive labeled and unlabeled data. However, the subsequent study indicates that uPU may be overfitting because its negative-class empirical risk estimator tends to be less than 0 during model training, so there is a non-negativity constraint, upgrading uPU to nnPU [12]. In addition, some variants replace the non-negativity constraint with other tricks such as absolute-value correction [15].

In this paper, we review some intriguing properties of nnPU since it is an efficient benchmark risk estimator in PU learning. We raise a question about **imbalance classification error rates in nnPU**: the negative-class empirical risk estimator approaches 0, whether the negative class is over-played even with the non-negativity constraint, resulting in imbalanced treatment between positive and negative classes. To respond to this question, we evaluate nnPU by designing certain early experiments, and empirical observations suggest that the answer to the question is YES (see details in **Sec 2.2**).

To upgrade nnPU, we suggest a dual-constrained risk estimator of PU learning named **DC-PU**, which applies two straightforward revisions. To maintain balanced error rates between the positive and negative classes, we suppose that the expected risks of the positive-class and negative-class should be close. Accordingly, we constrain that the negative-class empirical risk estimator is lower bounded by the positive-class empirical risk, instead of 0, *i.e.* the non-negativity constraint; and also incorporate an explicit equality constraint between them. We suggest an efficient training method for DC-PU based on the augmented Lagrange multiplier framework. We theoretically analyze the estimation error of DC-PU. We also empirically evaluate DC-PU on benchmark PU learning datasets. The results demonstrate that (1) compared with other risk estimators of PU learning, DC-PU achieves higher accuracy and converges more stably, and (2) compared with practical PU learning methods, DC-PU performs competitive accuracy performance.

- We empirically evaluate the problem of balancing positive and negative classification error rates in the risk estimator nnPU.
- To solve the problem, we propose a novel risk estimator of PU learning named DC-PU that balances classification error rates, and theoretically analyze it.
- We conduct extensive experiments to indicate the effectiveness of DC-PU.

## 2 Preliminaries and Analysis

In this section, we briefly review the preliminaries of PU learning; and then empirically investigate the problem of balancing classification error rates in PU learning risk estimator.

### 2.1 Preliminaries of PU learning

**Problem setup** Formally, let $\mathbf{x} \in \mathcal{X} \subset \mathbb{R}^d$ and $y \in \mathcal{Y} = \{-1, +1\}$ denote a $d$-dimensional input feature and a binary label, respectively. PN learning is the standard supervised binary classification trained using positive labeled data $\mathcal{D}_p$ and negative labeled data $\mathcal{D}_u$:

$$\mathcal{D}_p = \left\{ (\mathbf{x}_i^p, +1) \mid \mathbf{x}_i^p \overset{i.i.d.}{\sim} p(\mathbf{x}|y = +1) \right\}_{i=1}^{n_p}$$
$$\mathcal{D}_n = \left\{ (\mathbf{x}_i^n, -1) \mid \mathbf{x}_i^n \overset{i.i.d.}{\sim} p(\mathbf{x}|y = -1) \right\}_{i=1}^{n_n},$$

where $n_p$ and $n_n$ represent the numbers of the positive and negative labeled samples, respectively, and each sample is drawn independently and identically distributed from its corresponding class conditional distribution.

PU learning is a special weakly-supervised binary classification trained using positive labeled data $\mathcal{D}_p$ and unlabeled data $\mathcal{D}_u$. In this work, we concentrate on the *two-sample problem setting* [14]. The *completely selected at random* assumption [16] is applied, so $\mathcal{D}_p$ and $\mathcal{D}_u$ are drawn independently

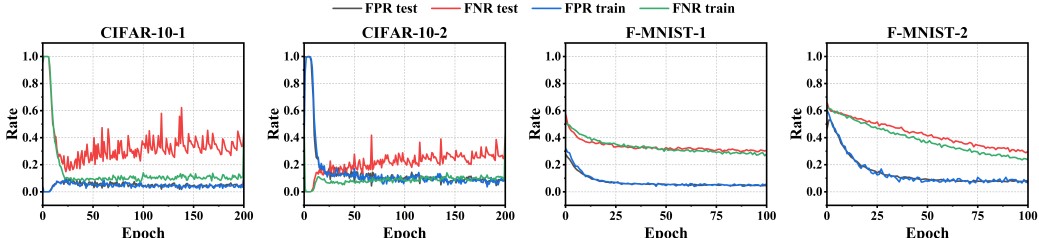

Figure 1: False positive rate and false negative rate on 4 benchmark PU learning datasets.

and identically distributed from the positive-class conditional distribution and the whole population:

$$\mathcal{D}_p = \left\{ (\mathbf{x}_i^p, +1) \mid \mathbf{x}_i^p \overset{i.i.d.}{\sim} p(\mathbf{x}|y=+1) \right\}_{i=1}^{n_p}$$

$$\mathcal{D}_u = \left\{ (\mathbf{x}_i^u, y_i^u) \mid \mathbf{x}_i^u \overset{i.i.d.}{\sim} p(\mathbf{x}) \right\}_{i=1}^{n_u},$$

where $n_u$ denotes the number of unlabeled samples.

**Risk estimators**  The objective is to induce a classifier $g : \mathcal{X} \to \mathcal{Y}$ that can predict the labels for any future samples. Let $\ell : \mathcal{X} \times \mathcal{Y} \to \mathbb{R}_+$ be a loss function such as cross-entropy. In PN learning, its expected risk can be formulated as follows:

$$R(g) = \pi R_p^+(g) + (1-\pi)R_n^-(g), \tag{1}$$

where $\pi$ represents the positive class prior $p(y = +1)$; $R_p^+(g) := \mathbb{E}_{\mathcal{X} \sim p(\mathbf{x}|y=+1)} \left[ \ell\big(g(\mathbf{x}), +1\big) \right]$ and $R_n^-(g) := \mathbb{E}_{\mathcal{X} \sim p(\mathbf{x}|y=-1)} \left[ \ell\big(g(\mathbf{x}), -1\big) \right]$. Given $\mathcal{D}_p \cup \mathcal{D}_n$, the approximation of Eq.(1), *i.e.* the empirical risk, is given below:

$$\widehat{R}_{\text{PN}}(g) = \pi \widehat{R}_p^+(g) + (1-\pi)\widehat{R}_n^-(g) \tag{2}$$

where $\widehat{R}_p^+(g) := \frac{1}{n_p}\sum_{i=1}^{n_p} \left[ \ell\big(g(\mathbf{x}_i^p), +1\big) \right]$ and $\widehat{R}_n^-(g) := \frac{1}{n_n}\sum_{i=1}^{n_n} \left[ \ell\big(g(\mathbf{x}_i^n), -1\big) \right]$.

In PU learning, the negative labeled data are unavailable, so its corresponding risk $R_n^-(g)$ can be optimized by no means. Thanks to the *completely selected at random* assumption [16], we have $p(\mathbf{x}) = \pi p(\mathbf{x}|y = +1) + (1-\pi)p(\mathbf{x}|y = -1)$, leading to $\pi R_n^-(g) = R_u^-(g) - \pi R_p^-(g)$, where $R_p^-(g) := \mathbb{E}_{\mathcal{X} \sim p(\mathbf{x}|y=+1)} \left[ \ell\big(g(\mathbf{x}), -1\big) \right]$. Accordingly, given $\mathcal{D}_p \cup \mathcal{D}_u$, the expected risk $R(g)$ of Eq.1 can be approximated by an unbiased empirical risk for PU learning (uPU) [11]:

$$\widehat{R}_{\text{uPU}}(g) = \pi \widehat{R}_p^+(g) + \widehat{R}_u^-(g) - \pi \widehat{R}_p^-(g) \tag{3}$$

where $\widehat{R}_u^-(g) := \frac{1}{n_u}\sum_{i=1}^{n_u} \left[ \ell\big(g(\mathbf{x}_i^u), -1\big) \right]$ and $\widehat{R}_p^-(g) := \frac{1}{n_p}\sum_{i=1}^{n_p} \left[ \ell\big(g(\mathbf{x}_i^p), -1\big) \right]$.

Unfortunately, uPU suffers from severe overfitting because the term $\widehat{R}_u^-(g) - \pi\widehat{R}_p^-(g)$ often continues to decrease and goes negative during model training. To alleviate this problem, [12] proposes a non-negativity constraint on this term, leading to a non-negative risk estimator for PU learning (nnPU) formulated as follows:

$$\widehat{R}_{\text{nnPU}}(g) = \pi \widehat{R}_p^+(g) + \max\{0, \widehat{R}_u^-(g) - \pi\widehat{R}_p^-(g)\} \tag{4}$$

## 2.2   Empirical Observations of nnPU

We revisit nnPU that the term $\widehat{R}_u^-(g) - \pi\widehat{R}_p^-(g)$ is used to approximate $(1-\pi)R_n^-$, however, it often continues to rapidly decrease and be truncated by 0 with the non-negativity constraint [12]. Therefore, we raise the question of whether the negative class is over-played, potentially resulting in imbalance between generalization errors of the positive-class $R_p^+$ and the negative-class $R_n^-$. To answer this question, we empirically measure $R_p^+$ by **F**alse **N**egative **R**ate (**FNR**) and $R_n^-$ by **F**alse **P**ositive **R**ate

(**FPR**) given available labeled data $\{(\mathbf{x}_i, y_i)\}_{i=1}^n$:

$$R_p^+ \approx \text{FNR} = \frac{\sum_{i=1}^n \mathbb{I}(y_i = +1 \wedge g(\mathbf{x}_i) = -1)}{\sum_{i=1}^n \mathbb{I}(y_i = +1)}$$

$$R_n^- \approx \text{FPR} = \frac{\sum_{i=1}^n \mathbb{I}(y_i = -1 \wedge g(\mathbf{x}_i) = +1)}{\sum_{i=1}^n \mathbb{I}(y_i = -1)}, \tag{5}$$

where $\mathbb{I}(\cdot)$ denotes the indicator function.

We conducted early experiments on 4 benchmark PU learning datasets (see details in **Sec 5.1**). Specifically, for each dataset, we apply nnPU to train a classifier $g$ and report the scores of FNR and FPR measured on both the training and test sets. As depicted in Fig.1, we can observe that the scores of FPR are consistently lower than those of FNR in all the settings, and the gaps between them are significantly large in some cases. For example, in terms of CIFAR-10-1 and F-MNIST-2, the scores of FPR are about $0.3 \sim 0.4$ and $0.2 \sim 0.3$ lower than those of FNR on the test sets. These empirical results validate that the classifier trained by nnPU tends to predict more accurately for the negative-class than the positive-class, resulting in the **imbalanced classification error rates problem**. This phenomenon is consistent to the previous finding that the term $\widehat{R}_u^-(g) - \pi \widehat{R}_p^-(g)$ of nnPU often approaches 0 [12], so as to over-play the negative-class.

## 3 Balancing Classification Error Rates in PU Learning

In this section, we propose a PU risk estimator of PU learning named DC-PU, which balances positive and negative classification error rates.

### 3.1 DC-PU PU Risk Estimator

We empirically validate that nnPU may result in the imbalance problem between $R_p^+$ and $R_n^-$ because the empirical estimator of $R_n^-$ is lower than that of $R_p^+$. To solve this problem, in DC-PU we suppose that $R_p^+$ and $R_n^-$ should be close, so we have two equality constraints on their empirical estimators.

Specifically, we revisit the empirical estimators of $R_p^+$ and $R_n^-$ in nnPU as follows:

$$R_p^+ \approx \widehat{R}_p^+; \quad R_n^- \approx \frac{(\widehat{R}_u^-(g) - \pi \widehat{R}_p^-(g))}{1 - \pi}, \tag{6}$$

and have **weak** and **strong** equality constraints on them. **First**, the weak constraint is that $\frac{(\widehat{R}_u^-(g) - \pi \widehat{R}_p^-(g))}{1 - \pi}$ is dynamically lower bounded by the fixed state of $\widehat{R}_p^+$, denoted by $\omega$, rather than the non-negativity constraint in nnPU. Accordingly, we can reformulate the empirical risk as follows:

$$\widehat{R}_{\text{DC-PU}}(g) = \pi \widehat{R}_p^+(g) + \max \left\{ \omega, \widehat{R}_u^-(g) - \pi \widehat{R}_p^-(g) \right\} \tag{7}$$

where $\omega$ is dynamically updated during classifier training. **Second**, the strong constraint is an explicit equality constraint. Upon these ideas, our DC-PU is finally formulated as follows:

$$\min_g \ \pi \widehat{R}_p^+(g) + \max \left\{ \omega, \widehat{R}_u^-(g) - \pi \widehat{R}_p^-(g) \right\} \quad \text{s.t.} \ \widehat{R}_p^+(g) = \frac{\widehat{R}_u^-(g) - \pi \widehat{R}_p^-(g)}{1 - \pi} \tag{8}$$

**Training** DC-PU is intractable to optimize because it involves an explicit equality constraint. To this end, we suggest an efficient training method with the augmented Lagrange multiplier framework. Following [17], we can transform Eq.(8) into the following augmented Lagrange problem:

$$\min_{g,\Theta} \ \pi \widehat{R}_p^+(g) + \max \left\{ \omega, \widehat{R}_u^-(g) - \pi \widehat{R}_p^-(g) \right\} + \frac{\tau}{2} \left\| \widehat{R}_p^+(g) - \frac{\widehat{R}_u^-(g) - \pi \widehat{R}_p^-(g)}{1 - \pi} + \frac{\Theta}{\tau} \right\|_2^2, \tag{9}$$

where $\| \cdot \|_2$ denotes the $\ell_2$ norm; $\Theta$ is the Lagrange parameter; and $\tau$ is the penalty parameter. Accordingly, we can directly apply any gradient-based method to optimize Eq.(9) with respect to $\{g, \Theta\}$.

---

**Algorithm 1** Training of DC-PU

---

**Input:** PU learning dataset $\mathcal{D}_p \cup \mathcal{D}_u$; method parameters $\beta$, $\tau$, $\gamma$; number of iterations $T$.
**Output:** A trained classifier $g$.
1: Initialize $g$ with pre-trained backbone and $\Theta$ randomly
2: **for** $t = 1, 2, \ldots, T$ **do**
3:    Draw a mini-batch $\mathcal{D}_p^{(t)} \cup \mathcal{D}_u^{(t)}$ from $\mathcal{D}_p \cup \mathcal{D}_u$
4:    Compute $\widehat{R}_p^+(g; \mathcal{D}_p^{(t)})$, $\widehat{R}_u^-(g; \mathcal{D}_u^{(t)})$, and $\widehat{R}_p^-(g; \mathcal{D}_p^{(t)})$
5:    Update $\omega^{(t)}$ by using Eq.(10)
6:    **if** $\widehat{R}_u^-(g; \mathcal{D}_u^{(t)}) - \pi \widehat{R}_p^-(g; \mathcal{D}_p^{(t)}) \geq \omega^{(t)} - \gamma$ **then**
7:       Compute the stochastic gradient $\nabla_g \mathcal{L}$
8:    **else**
9:       Compute the gradient $\nabla_g \mathcal{L}$ as $\nabla_g \left( \widehat{R}_u^-(g; \mathcal{D}_u^{(t)}) - \pi \widehat{R}_p^-(g; \mathcal{D}_p^{(t)}) \right)$
10:   **end if**
11:   Update $g$ with $\nabla_g \mathcal{L}$ and any adaptive learning rate method
12:   Update $\Theta$ by using Eq.(11)
13: **end for**

---

To handle large-scale data, we employ the stochastic gradient decent method. At each iteration $t$, we draw a mini-batch $\mathcal{D}_p^{(t)} \cup \mathcal{D}_u^{(t)}$ from $\mathcal{D}_p \cup \mathcal{D}_u$, and compute $\widehat{R}_p^+(g; \mathcal{D}_p^{(t)})$, $\widehat{R}_u^-(g; \mathcal{D}_u^{(t)})$, and $\widehat{R}_p^-(g; \mathcal{D}_p^{(t)})$. In terms of $\omega$, we update it with an exponential moving average trick:

$$\omega^{(t)} = \beta \omega^{(t-1)} + (1 - \beta)(1 - \pi) \widehat{R}_p^+(g; \mathcal{D}_p^{(t)}), \tag{10}$$

where $\beta$ is the moving parameter. In terms of $g$, we update it with the stochastic gradient formed by $\mathcal{D}_p^{(t)} \cup \mathcal{D}_u^{(t)}$ and solve the max operator by the rollback strategy suggested by [12]. In terms of $\Theta$, it can be updated as follows:

$$\Theta^{(t)} = \Theta^{(t-1)} + \tau \left( \widehat{R}_p^+(g; \mathcal{D}_p^{(t)}) - \frac{\widehat{R}_u^-(g; \mathcal{D}_u^{(t)}) - \pi \widehat{R}_p^-(g; \mathcal{D}_p^{(t)})}{1 - \pi} \right) \tag{11}$$

For clarity, we summarize the full training process of DC-PU in **Algorithm 1**.

### 3.2 Theoretical Analysis

In this section, we analyze the bias, consistency, and estimation error of DC-PU given in Eq.(8) (all proofs are in Appendix A).

We first clarify some symbols for the following analyses. Let $\mathcal{G}$ be a hypothesis space that satisfies the boundedness property, characterized by the existence of a constant $C_g > 0$ ensuring $\sup_{g \in \mathcal{G}} \|g\|_\infty \leq C_g$ and closure under negation. On this space, we define a loss function $\ell : \mathbb{R} \times \pm 1 \to \mathbb{R}+$ that exhibits $L_\ell$-Lipschitz continuity and is bounded by a constant $C_\ell > 0$, such that $\ell(t, y) \leq C_\ell$ holds for all $|t| \leq C_g$ and $y \in \pm 1$. To ensure the learning feasibility, we impose three additional conditions: a separability condition requiring the existence of $\alpha > 0$ such that $\inf_{g \in \mathcal{G}} R_n^-(g) \geq \alpha$, the upper bound condition requiring the existence of $\beta > 0$ such that $0 \leq \sup_{g \in \mathcal{G}} \widehat{R}_p^+(g) \leq \beta$, and a complexity condition constraining the Rademacher complexity of $\mathcal{G}$ to satisfy $\mathcal{R}_p(\mathcal{G}) = \mathcal{O}(1/\sqrt{n_p})$ and $\mathcal{R}_u(\mathcal{G}) = \mathcal{O}(1/\sqrt{n_u})$. With the DC-PU risk of Eq.(8), we can partition all possible $(\mathcal{D}_p, \mathcal{D}_u)$ into $\mathfrak{D}_\omega^+ = \{(\mathcal{D}_p, \mathcal{D}_u) \mid \widehat{R}_u^-(g) - \pi \widehat{R}_p^-(g) \geq \omega\}$ and $\mathfrak{D}_\omega^- = \{(\mathcal{D}_p, \mathcal{D}_u) \mid \widehat{R}_u^-(g) - \pi \widehat{R}_p^-(g) < \omega\}$ where $\omega$ is decided by $(1 - \pi) \widehat{R}_p^+(g)$.

**Lemma 3.1.** $\widehat{R}_{\text{DC-PU}}(g)$ is positively biased from $\widehat{R}(g)$ with a non-zero probability $\mathcal{P}(\mathfrak{D}_\omega^-(g))$ over repeated sampling of $(\mathcal{D}_p, \mathcal{D}_u)$, which can be bounded by

$$\mathcal{P}(\mathfrak{D}_\omega^-(g)) \leq \exp \left( -\frac{2(1 - \pi)^2 (\alpha - \beta)^2 / C_\ell^2}{\pi^2 / n_p + 1 / n_u} \right). \tag{12}$$

Based Lemma 3.1, we can present the exponential decay of the bias and also the consistency for the DC-PU risk of Eq.(8) with the following theorem.

**Theorem 3.2.** *Denote the bound of $\mathcal{P}(\mathfrak{D}_\omega^-(g))$ in Eq.(12) by $\Delta$, $\pi' = \max(\pi, 1 - \pi)$ and $\chi_{n_p, n_u} = 2\pi/\sqrt{n_p} + 1/\sqrt{n_u}$. It holds that*

$$0 \le \mathbb{E}_{\mathcal{D}_p, \mathcal{D}_u}[\widehat{R}_{\text{DC-PU}}(g)] - R(g) \le \pi' C_\ell \Delta \tag{13}$$

*For any $\delta > 0$, it has with probability at least $1 - \delta$*

$$|\widehat{R}_{\text{DC-PU}}(g) - R(g)| \le C_\delta \chi_{n_p, n_u} + \pi' C_\ell \Delta \tag{14}$$

*and with probability at least $1 - \delta - \Delta$*

$$|\widehat{R}_{\text{DC-PU}}(g) - R(g)| \le C_\delta \chi_{n_p, n_u} \tag{15}$$

*where $C_\delta = C_\ell \sqrt{\ln(2/\delta)/2}$.*

Theorem 3.2 indicates that for fixed $g$, $\widehat{R}_{\text{DC-PU}}(g) \to R(g)$ with a convergence rate $\mathcal{O}_p(\pi/\sqrt{n_p} + 1/\sqrt{n_u})$, which is optimal according to the central limit theorem. It means that the proposed DC-PU risk in Eq.(8) is a biased yet optimal estimator to the PN risk.

**Theorem 3.3.** *Let $g^* = \arg\min_{g \in \mathcal{G}} R(g)$ is the minimizer of the true classification risk in Eq.2 and $\hat{g}_{\text{DC-PU}} = \arg\min_{g \in \mathcal{G}} \hat{R}_{\text{DC-PU}}(g)$ denotes the minimizer of the risk form in Eq.8. Denote the bound of $\mathcal{P}(\mathfrak{D}_\omega^-(g))$ in Eq.(12) by $\Delta$, $\pi' = \max(\pi, 1 - \pi)$ and $\chi_{n_p, n_u} = 2\pi/\sqrt{n_p} + 1/\sqrt{n_u}$. Then for any $\delta > 0$, it holds with probability at least $1 - \delta$:*

$$R(\hat{g}_{\text{DC-PU}}) - R(g^*) \le 16 L_\ell \mathfrak{R}_{n_p, p_p}(\mathcal{G}) + 8 L_\ell \mathfrak{R}_{n_u, p}(\mathcal{G}) + 2 C'_\delta \chi_{n_p, n_u} + 2\pi' C_\ell \Delta \tag{16}$$

*where $C'_\delta = C_\ell \sqrt{\ln(1/\delta)/2}$.*

Theorem 3.3 establishes the fundamental generalization bound that explains DC-PU's superior performance through four distinct components: two Rademacher complexity terms $\mathfrak{R}_{n_u, p}(\mathcal{G})$ and $\mathfrak{R}_{n_p, p_p}(\mathcal{G})$, a sampling fluctuation term $\chi_{n_p, n_u}$, and a distribution discrepancy term $\Delta$. Under standard assumptions, as $n_p, n_u \to \infty$, two Rademacher complexity terms gradually decrease and converge to zero, and so does the sampling fluctuation term $\chi_{n_p, n_u}$. The bias term $\Delta$ is independent of sample size and can be treated as a constant. Consequently, as $n \to \infty$, $R(\hat{g}_{\text{DC-PU}}) \to R(g^*)$. Additionally, the convergence rates of the first and second terms are governed by the Rademacher complexities $\mathfrak{R}_{n_u, p}(\mathcal{G})$ and $\mathfrak{R}_{n_p, p_p}(\mathcal{G})$, corresponding to rates of $O(1/\sqrt{n_p})$ and $O(1/\sqrt{n_u})$, respectively. The fluctuation term $\chi_{n_p, n_u}$ contributes an additional rate of $O(1/\sqrt{n_p} + 1/\sqrt{n_u})$. Consequently, the overall convergence rate is characterized by $O\left(\max(1/\sqrt{n_p}, 1/\sqrt{n_u})\right)$. It indicates that DC-PU achieves minimax optimal convergence while maintaining balanced performance, contrasting with methods that optimize overall accuracy at the expense of class balance.

## 4 Related Works

Recent years have witnessed significant advancements in PU learning, with research on risk estimators playing a pivotal role in its theoretical foundation. uPU [11] presents a solution to PU learning problems by constructing an unbiased risk estimator, which lays the theoretical foundation for future research. However, when models become complex, the unbiased risk estimator produces negative empirical risks, resulting in overfitting problems. To address this, nnPU [12] reduces overfitting by adding non-negative constraints on negative risks. Following this, abs-PU [15] introduces a simplified method using absolute value correction to handle non-negative constraints. Building on these works, Dist-PU [13] adopt a novel perspective by aligning predicted and ground-truth label distributions to address negative prediction bias. A recent contribution, FOPU [18], extend the risk estimator framework by incorporating fairness constraints to address group fairness problems in text classification while maintaining model performance. Handling imperfect annotations has also been studied in related weakly-supervised settings [19, 20, 21, 22].

Beyond risk estimator-based approaches, researchers explore various strategies in PU learning. Sample selection methods focus on identifying reliable negative instances from unlabeled data, exemplified by the probabilistic estimation approach in PUbN [23] and the dynamic weight adjustment mechanism in Robust-PU [24]. The recent VQ-Encoder [25] learns disentangled representations

Table 1: Detailed characteristics of datasets.

| Dataset | #Input | #Train | #Test | $\pi$ | Positive Class | Backbone |
|---------|--------|--------|-------|-------|----------------|----------|
| F-MNIST-1 | $28 \times 28$ | 60,000 | 10,000 | 0.3 | 1,4,7 | 5-layer MLP |
| F-MNIST-2 | $28 \times 28$ | 60,000 | 10,000 | 0.7 | 0,2,3,5,6,8,9 | 5-layer MLP |
| CIFAR-10-1 | $3 \times 32 \times 32$ | 50,000 | 10,000 | 0.4 | 0,1,8,9 | 12-layer CNN |
| CIFAR-10-2 | $3 \times 32 \times 32$ | 50,000 | 10,000 | 0.6 | 2,3,4,5,6,7 | 12-layer CNN |
| Alzheimer | $3 \times 224 \times 224$ | 5,121 | 1,279 | 0.499 | 0,1,3 | ResNet-50 |

from a representation learning perspective to achieve clustering separation of unlabeled data. Another category comprises generation-based methods, such as GenPU [26], which utilizes generative adversarial networks to synthesize negative samples. Recent studies also integrate self-supervised [27] and predictive trend detection [8] frameworks to enhance feature representations.

In contrast to these approaches, our method specifically addresses the problem of excessive negative class emphasis in PU learning, which can result in disparate treatment between positive and negative classes. Accordingly, we propose DC-PU, a balanced and robust risk estimator.

# 5 Experiments

In this section, we compare the performance of DC-PU with existing risk estimators on commonly used benchmarks.

## 5.1 Settings

**Datasets** To comprehensively evaluate the proposed method, we conduct experiments on two widely-adopted benchmark datasets: Fashion-MNIST (F-MNIST)[28] and CIFAR-10[29], along with a real-world medical dataset Alzheimer. Based on the characteristics of different datasets, we design corresponding model architectures as backbone networks following nnPU [12]. Given that these datasets are originally designed for multi-class classification tasks, we transform them into binary classification problems by partitioning the class labels into positive and negative categories. Table 1 presents detailed statistics of each dataset.

**Baseline methods** To evaluate the balance of classification error rates in our proposed risk estimator, we select five classical risk estimator methods for comparison: uPU [11], nnPU [12], abs-PU [15], Dist-PU [13], and FOPU [18]. Given that our focus is on the assessment at the risk estimator level, we simplify the regularization terms in Dist-PU and FOPU to construct their baseline versions, Dist-PU* and FOPU*. The details of baselines are presented in the Appendix B. All experiments are conducted on a server equipped with two Nvidia RTX4090 GPUs.

**Evaluation metrics** Here, we measure the classification performance by employing Micro-F1 and Macro-F1, where higher scores indicate better performance. Besides, we measure the balance performance by employing the GAP between FNR and FPR, formulated as follows:

$$\text{GAP} = |\text{FNR} - \text{FPR}| \tag{17}$$

For the GAP metric, lower scores imply better performance.

## 5.2 Comparing with Existing Risk Estimators

From the perspective of error rates, the GAP metrics delineated in Fig.2 provide crucial insights into the performance of different PU risk estimators in balancing classification error rates. The GAP between FPR and FNR demonstrates that DC-PU consistently maintains lower values across all four benchmark datasets (CIFAR-10-1, CIFAR-10-2, F-MNIST-1, and F-MNIST-2), indicating superior performance in balancing classification error rates. For example, in the F-MNIST-1 dataset, DC-PU (denoted by the yellow line) maintains consistently low GAP values throughout the training process, ultimately converging to approximately 0.05, whereas methods such as nnPU exhibit notably higher GAP values. This exceptional performance can be attributed to the explicit equality constraint in DC-PU's objective function (specifically, $\widehat{R}_p^+(g) = \frac{\widehat{R}_u^-(g) - \pi \widehat{R}_p^-(g)}{1-\pi}$), which actively promotes

Table 2: Results of classification evaluation metrics (mean±std) on four benchmark PU datasets, where "max" represents the maximum value achieved during training and "last" indicates the average performance over the final 5 epochs. The highest scores are indicated in **bold**.

| Dataset | Method | Micro-F1 | | Macro-F1 | |
|---|---|---|---|---|---|
| | | max | last | max | last |
| CIFAR-10-1 | uPU | 86.61±0.004 | 61.19±0.000 | 85.91±0.005 | 40.94±0.000 |
| | nnPU | 88.72±0.001 | 86.29±0.001 | 88.06±0.001 | 85.68±0.002 |
| | abs-PU | 88.80±0.002 | 85.86±0.006 | 88.27±0.003 | 85.29±0.007 |
| | Dist-PU* | 89.14±0.000 | 86.43±0.007 | 88.73±0.000 | 86.00±0.007 |
| | FOPU* | 88.49±0.001 | 82.53±0.002 | 87.88±0.002 | 82.09±0.002 |
| | **Dc-PU** | **89.37±0.001** | **88.46±0.007** | **88.84±0.001** | **88.11±0.006** |
| CIFAR-10-2 | uPU | 87.53±0.007 | 41.62±0.001 | 86.98±0.007 | 31.79±0.003 |
| | nnPU | 88.10±0.001 | 85.26±0.01 | 87.42±0.001 | 84.51±0.011 |
| | abs-PU | 88.59±0.000 | 85.29±0.005 | 87.97±0.001 | 84.76±0.004 |
| | Dist-PU* | 88.23±0.001 | 85.15±0.002 | 87.58±0.001 | 84.56±0.002 |
| | FOPU* | 87.83±0.001 | 82.53±0.005 | 87.41±0.001 | 82.09±0.005 |
| | **Dc-PU** | **88.60±0.001** | **87.03±0.003** | **88.02±0.001** | **86.53±0.004** |
| F-MNIST-1 | uPU | 90.90±0.002 | 74.14±0.002 | 88.81±0.004 | 54.85±0.012 |
| | nnPU | 90.92±0.000 | 88.06±0.001 | 89.17±0.000 | 85.79±0.002 |
| | abs-PU | 90.87±0.000 | 87.83±0.001 | 88.74±0.001 | 85.74±0.001 |
| | Dist-PU* | 91.25±0.000 | 87.98±0.005 | 89.27±0.000 | 85.68±0.004 |
| | FOPU* | 90.91±0.000 | 88.50±0.004 | 89.03±0.001 | 85.82±0.007 |
| | **Dc-PU** | **91.48±0.001** | **89.97±0.001** | **89.86±0.001** | **88.32±0.002** |
| F-MNIST-2 | uPU | 85.57±0.011 | 49.49±0.084 | 83.09±0.009 | 48.98±0.092 |
| | nnPU | 88.53±0.002 | 83.79±0.007 | 85.78±0.002 | 81.10±0.009 |
| | abs-PU | 88.71±0.002 | 83.50±0.001 | 86.12±0.002 | 81.46±0.003 |
| | Dist-PU* | 88.40±0.011 | 84.65±0.007 | 86.41±0.015 | 81.68±0.011 |
| | FOPU* | 88.14±0.003 | 84.41±0.010 | 85.83±0.006 | 81.86±0.011 |
| | **Dc-PU** | **88.91±0.003** | **86.44±0.007** | **86.72±0.006** | **83.43±0.002** |

Table 3: Results of Micro-F1, Macro-F1 and GAP (mean±std) on Alzheimer, where "min" represents the minimum value achieved during training. The highest scores are indicated in **bold**.

| Method | Micro-F1 | | Macro-F1 | | GAP | |
|---|---|---|---|---|---|---|
| | max | last | max | last | min | last |
| uPU | 71.02±0.004 | 63.73±0.096 | 70.93±0.005 | 61.94±0.262 | 0.59±0.000 | 17.55±1.346 |
| nnPU | 70.23±0.002 | 67.87±0.055 | 68.98±0.002 | 66.61±0.203 | 0.46±0.000 | 16.49±0.224 |
| abs-PU | 71.09±0.002 | 67.92±0.003 | 70.79±0.001 | 67.00±0.010 | 0.44±0.003 | 13.44±0.695 |
| Dist-PU* | 71.25±0.002 | 68.56±0.046 | 71.14±0.001 | 66.94±0.116 | 0.37±0.001 | 13.65±0.383 |
| FOPU* | **71.56±0.002** | 64.14±0.053 | **71.27±0.002** | 60.96±0.172 | 0.26±0.000 | 12.57±0.706 |
| **Dc-PU** | 71.48±0.002 | **70.55±0.002** | 71.03±0.001 | **69.99±0.009** | **0.09±0.000** | **10.79±0.294** |

balanced prediction between positive and negative classes. This observation is further substantiated by the results presented in Table 3, where Dc-PU achieves significantly lower GAP values (0.009 at "min" and 10.79 at "last") compared to nnPU (0.46 at "min" and 16.49 at "last") on the Alzheimer dataset.

From the perspective of the convergence, Fig.3 and Fig.4 present the Micro-F1 and Macro-F1 performance trends during the training process across four benchmark datasets, respectively. Dc-PU exhibits outstanding convergence stability and final performance, as demonstrated by its consistently higher Micor-F1 and Macro-F1 values compared to other PU risk estimator. This pattern is particularly evident in the CIFAR-10-1 and F-MNIST-1 datasets, where Dc-PU maintains a stable high F1 level. In contrast, uPU shows significant stability issues, characterized by sharp F1 scores decline following initial convergenc. This instability can be attributed to the unconstrained nature of uPU's basic risk formulation, which fails to maintain consistent performance throughout the training process. The improved stability observed in newer methods such as nnPU and abs-PU can be attributed to their

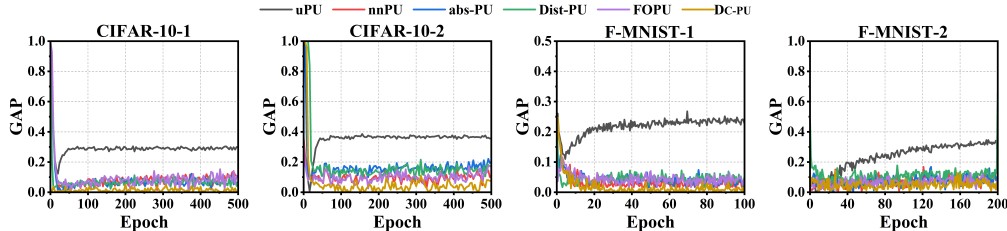

Figure 2: The curve of GAP during the training process on 4 benchmark PU learning datasets.

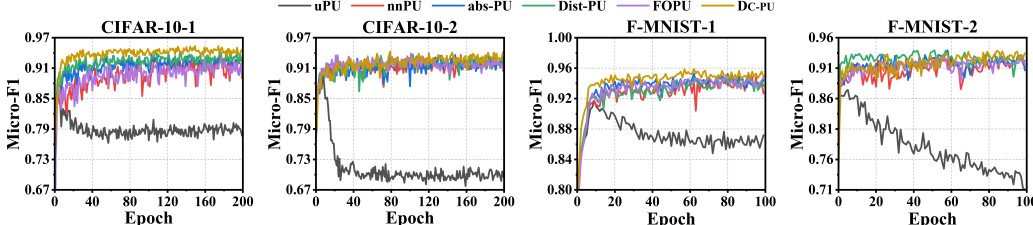

Figure 3: The curve of Micro-F1 during the training process on 4 benchmark PU learning datasets.

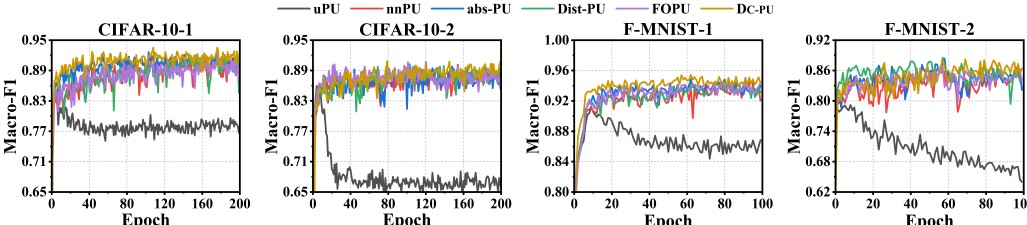

Figure 4: The curve of Macro-F1 during the training process on 4 benchmark PU learning datasets.

enhanced risk estimators, which incorporate non-negative constraints or absolute value corrections, effectively preventing model overfitting. Moreover, the results presented in Table 2 further substantiate the convergence advantages, with DC-PU consistently achieving superior Micro-F1 and Macro-F1 scores across all four datasets, both in terms of "last" and "max" performance metrics. Specifically, on the CIFAR-10-1 dataset, DC-PU attains a maximum Micro-F1 of 0.892 and final Micro-F1 of 0.875, surpassing the next best performer (nnPU) by margins of 0.043 and 0.038 respectively. Similarly, for Macro-F1 scores, DC-PU demonstrates exceptional performance on F-MNIST-2, achieving 0.901 at "max" and 0.889 at "last", while competing methods such as uPU and nnPU only reach maximum values of 0.847 and 0.862 respectively. This performance differential is particularly pronounced in the CIFAR-10-2 dataset, where DC-PU maintains consistently superior metrics with a maximum Micro-F1 of 0.903 and Macro-F1 of 0.897, representing improvements of approximately 5.2% and 4.8% over the baseline methods.

To further validate the effectiveness of our proposed method, we incorporated the additional regularization term inspired by Dist-PU into the aforementioned five risk estimators for comparison as well as our risk estimator. Due to the space limitation, the details are presented in the Appendix C.

### 5.3 Parameter Sensitivity

We conduct a comprehensive parameter sensitivity analysis with respect to the parameters $\tau$ and $\beta$, and the results are presented in Fig.5. For the parameter $\tau$, our experiments show optimal performance at $2 \times 10^{-3}$ across most datasets, which aligns with our theoretical analysis that moderate penalty parameters achieve balance between constraint strength and optimization stability. For the parameter $\beta$, we find that the range $[0.4, 0.5]$ performs optimally across different datasets, which is significant because $\beta$ controls the update speed of dynamic lower bound. Too small value of $\beta$ leads to overly loose constraints losing the balancing effect, while too large value of $\beta$ causes optimization instability. Our experiments demonstrate that DC-PU exhibits relative robustness to these hyperparameters,

Table 4: Results of ablative study (mean±std). The highest scores are indicated in **bold**.

| Variant | (a) | (b) | CIFAR-10-1 | CIFAR-10-2 | F-MNIST-1 | F-MNIST-2 | Alzheimer |
|---------|-----|-----|------------|------------|-----------|-----------|-----------|
| I | | | 88.72±0.001 | 88.10±0.001 | 90.92±0.000 | 88.53±0.002 | 70.23±0.002 |
| II | ✓ | | 89.25±0.001 | 88.42±0.000 | 91.43±0.000 | 88.56±0.001 | 70.94±0.001 |
| III | | ✓ | 88.90±0.001 | **88.60±0.001** | 91.21±0.000 | 88.88±0.001 | 71.33±0.006 |
| IV | ✓ | ✓ | **89.37±0.001** | **88.60±0.001** | **91.48±0.001** | **88.91±0.003** | **71.48±0.002** |

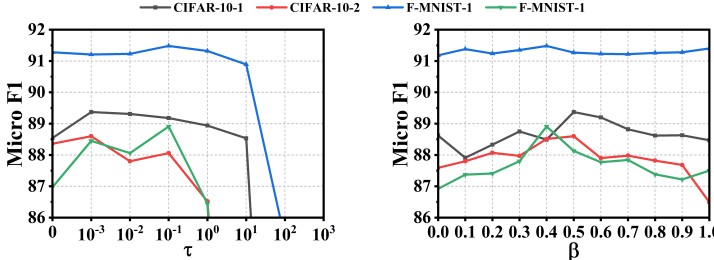

Figure 5: The parameter sensitivity analysis with respect to the parameters $\tau$ and $\beta$ on 4 benchmark PU learning datasets.

with parameter variations within reasonable ranges not significantly affecting performance, further validating the practicality of our method.

## 5.4 Ablation Study

We conduct ablation studies on five datasets to evaluate the impact of two key constraints: the weak constraint (a) and the strong constraint (b). Variants I-IV are designed to investigate the effects of these two constraints. The experimental results demonstrate that employing either the weak or strong constraint independently enhances model performance: Variant II with the weak constraint shows consistent improvements across all datasets, while Variant III with the strong constraint exhibits particularly notable performance on CIFAR-10-2 and F-MNIST-2. When both constraints are combined (Variant IV), the model achieves optimal performance, reaching 89.37% on CIFAR-10-1, 88.6% on CIFAR-10-2, 91.48% on F-MNIST-1, 88.91% on F-MNIST-2, and 71.48% on Alzheimer, which not only validates the complementary nature of these constraints but also substantiates the effectiveness of our proposed DC-PU.

## 6 Conclusion

In this paper, we observe a issue of balancing classification error rates in PU learning where the non-negativity constraint in nnPU leads to over-emphasis of the negative class. To address this, we propose DC-PU, a novel risk estimator that balances the error rates between positive and negative classes through two key constraints: a dynamic lower bound constraint and an explicit equality constraint. Through theoretical analysis and extensive experiments on benchmark datasets, DC-PU demonstrates improved evaluation metrics and stability across all datasets while balancing error rates between positive and negative classes compared to existing methods, establishing itself as an effective approach for achieving both high evaluation metrics and balanced classification error rates in PU learning.

## Acknowledgements

We would like to acknowledge support for this project from the National Science and Technology Major Project (No.2021ZD0112500), the National Natural Science Foundation of China (No.62276113), and the open research fund of Suzhou Key Lab of Multi-modal Data Fusion and Intelligent Healthcare.

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

# A   Proof of Theorems

## A.1   Proof of Lemma 3.1

*Proof.* Let $\mathcal{F}(\mathcal{D}_p), \mathcal{F}(\mathcal{D}_u)$ be the cumulative distribution function of $\mathcal{D}_p$ and $\mathcal{D}_u$, respectively, and $\mathcal{F}(\mathcal{D}_p, \mathcal{D}_u) = \mathcal{F}(\mathcal{D}_p) \cdot \mathcal{F}(\mathcal{D}_u)$ be the joint cumulative distribution function of $(\mathcal{D}_p, \mathcal{D}_u)$. Based on the unbiased fact of $\widehat{R}_{uPU}(g)$ and $\widehat{R}_{\text{DC-PU}}(g) - \widehat{R}_{uPU}(g) = 0$ on $\mathfrak{D}_\omega^+(g)$, it holds

$$
\begin{aligned}
\mathbb{E}[\widehat{R}_{\text{DC-PU}}(g)] - R(g) &= \mathbb{E}[\widehat{R}_{\text{DC-PU}}(g) - \widehat{R}_{uPU}(g)] \\
&= \int_{(\mathcal{D}_p, \mathcal{D}_u) \in \mathfrak{D}_\omega^+(g)} \widehat{R}_{\text{DC-PU}}(g) - \widehat{R}_{uPU}(g) d\mathcal{F}(\mathcal{D}_p, \mathcal{D}_u) \\
&\quad + \int_{(\mathcal{D}_p, \mathcal{D}_u) \in \mathfrak{D}_\omega^-(g)} \widehat{R}_{\text{DC-PU}}(g) - \widehat{R}_{uPU}(g) d\mathcal{F}(\mathcal{D}_p, \mathcal{D}_u) \\
&= \int_{(\mathcal{D}_p, \mathcal{D}_u) \in \mathfrak{D}_\omega^-(g)} \widehat{R}_{\text{DC-PU}}(g) - \widehat{R}_{uPU}(g) d\mathcal{F}(\mathcal{D}_p, \mathcal{D}_u).
\end{aligned}
$$

Due to the fact $\widehat{R}_{\text{DC-PU}}(g) - \widehat{R}_{uPU}(g) > 0$ on $\mathfrak{D}_\omega^-(g)$, $\mathbb{E}[\widehat{R}_{\text{DC-PU}}(g)] - R(g) > 0$ with the probability $\mathcal{P}(\mathfrak{D}_\omega^-(g)) = \int_{(\mathcal{D}_p, \mathcal{D}_u) \in \mathfrak{D}_\omega^-(g)} d\mathcal{F}(\mathcal{D}_p, \mathcal{D}_u)$. In other words, $\widehat{R}_{\text{DC-PU}}(g)$ is positively biased with the probability $\mathcal{P}(\mathfrak{D}_\omega^-(g))$. Given the assumptions $R_n^-(g) \geq \alpha > 0$ and $0 \leq \widehat{R}_p^+(g) \leq \beta$, we have

$$
\mathbb{E}[\widehat{R}_u^-(g) - \pi\widehat{R}_p^-(g)] = R_u^-(g) - \pi R_p^-(g) = (1-\pi)R_n^-(g) \geq (1-\pi)\alpha
$$

and

$$
\begin{aligned}
\mathcal{P}(\mathfrak{D}_\omega^-(g)) &= \mathcal{P}\left(\widehat{R}_u^-(g) - \pi\widehat{R}_p^-(g) < (1-\pi)\widehat{R}_p^+(g)\right) \\
&\leq \mathcal{P}\left(\widehat{R}_u^-(g) - \pi\widehat{R}_p^-(g) < (1-\pi)R_n^-(g) - (1-\pi)\alpha + (1-\pi)\widehat{R}_p^+(g)\right) \\
&\leq \mathcal{P}\left((1-\pi)R_n^-(g) - \left(\widehat{R}_u^-(g) - \pi\widehat{R}_p^-(g)\right) \geq (1-\pi)\alpha - (1-\pi)\widehat{R}_p^+(g)\right) \\
&\leq \mathcal{P}\left((1-\pi)R_n^-(g) - \left(\widehat{R}_u^-(g) - \pi\widehat{R}_p^-(g)\right) \geq (1-\pi)(\alpha - \beta)\right)
\end{aligned}
$$

According to the assumption $0 \leq \ell(t, \pm 1) \leq C_\ell$, the change of $\widehat{R}_p^-(g)$ will be no more than $C_\ell/n_p$ and the change of $\widehat{R}_u^-(g)$ no more than $C_\ell/n_u$ if we replace some $(\mathbf{x}_i^p, +1) \in \mathcal{D}_p$ or some $(\mathbf{x}_i^u, y_i^u) \in \mathcal{D}_u$. Subsequently, based on McDiarmid's inequality, it holds

$$
\begin{aligned}
\mathcal{P}\left((1-\pi)R_n^-(g) - \left(\widehat{R}_u^-(g) - \pi\widehat{R}_p^-(g)\right) \geq (1-\pi)(\alpha - \beta)\right) &\leq \exp\left(-\frac{2(1-\pi)^2(\alpha - \beta)^2}{n_p(C_\ell\pi_p/n_p)^2 + n_u(C_\ell/n_u)^2}\right) \\
&= \exp\left(-\frac{2(1-\pi)^2(\alpha - \beta)^2/C_\ell^2}{\pi^2/n_p + 1/n_u}\right).
\end{aligned}
$$

$\square$

## A.2   Proof of Theorem 3.2

*Proof.* Based on Lemma 3.1 and its proof, we can obtain the exponential decay of the bias in Eq.(13) by

$$
\begin{aligned}
\mathbb{E}_{\mathcal{D}_p, \mathcal{D}_u}[\widehat{R}_{\text{DC-PU}}(g)] - R(g) &\leq \sup_{(\mathcal{D}_p, \mathcal{D}_u) \in \mathfrak{D}_\omega^-(g)} (\widehat{R}_{\text{DC-PU}}(g) - \widehat{R}_{uPU}(g)) \cdot \int_{(\mathcal{D}_p, \mathcal{D}_u) \in \mathfrak{D}_\omega^-(g)} d\mathcal{F}(\mathcal{D}_p, \mathcal{D}_u) \\
&= \sup_{(\mathcal{D}_p, \mathcal{D}_u) \in \mathfrak{D}_\omega^-(g)} ((1-\pi)\widehat{R}_p^+(g) + \pi\widehat{R}_p^-(g) - \widehat{R}_u^-(g)) \cdot \mathcal{P}(\mathfrak{D}_\omega^-(g)) \\
&= \sup_{(\mathcal{D}_p, \mathcal{D}_u) \in \mathfrak{D}_\omega^-(g)} ((1-\pi)\widehat{R}_p^+(g) + \pi(C_\ell - \widehat{R}_p^+(g)) - \widehat{R}_u^-(g)) \cdot \mathcal{P}(\mathfrak{D}_\omega^-(g)) \\
&= \sup_{(\mathcal{D}_p, \mathcal{D}_u) \in \mathfrak{D}_\omega^-(g)} (\pi C_\ell + (1-2\pi)\widehat{R}_p^+(g) - \widehat{R}_u^-(g)) \cdot \mathcal{P}(\mathfrak{D}_\omega^-(g)) \\
&\leq \pi' C_\ell \Delta,
\end{aligned}
$$

where $\pi' = \max(\pi, 1 - \pi)$.

The deviation bound in Eq.(14) is obtained by:

$$\left|\widehat{R}_{\text{DC-PU}}(g) - R(g)\right| \leq \left|\widehat{R}_{\text{DC-PU}}(g) - \mathbb{E}[\widehat{R}_{\text{DC-PU}}(g)]\right| + \left|\mathbb{E}[\widehat{R}_{\text{DC-PU}}(g)] - R(g)\right|$$

$$\leq \left|\widehat{R}_{\text{DC-PU}}(g) - \mathbb{E}[\widehat{R}_{\text{DC-PU}}(g)]\right| + \pi' C_\ell \Delta.$$

According to the assumption $0 \leq \ell(t, \pm 1) \leq C_\ell$, the change of $\widehat{R}_{\text{DC-PU}}(g)$ will be no more than $2C_\ell/n_p$ or $C_\ell/n_u$ if we replace some $(\mathbf{x}_i^p, +1) \in \mathcal{D}_p$ or some $(\mathbf{x}_i^u, y_i^u) \in \mathcal{D}_u$. Subsequently, based on McDiarmid's inequality, it holds

$$\mathcal{P}(\left|\widehat{R}_{\text{DC-PU}}(g) - \mathbb{E}[\widehat{R}_{\text{DC-PU}}(g)]\right| \leq \epsilon) \leq 2\exp\left(-\frac{2\epsilon^2}{n_p(2C_\ell\pi/n_p) + n_u(C_\ell/n_u)}\right),$$

or equivalently, with probability at least $1 - \delta$

$$\left|\widehat{R}_{\text{DC-PU}}(g) - \mathbb{E}[\widehat{R}_{\text{DC-PU}}(g)]\right| \leq \sqrt{\frac{\ln(2/\delta)C_\ell^2}{2}\left(\frac{4\pi^2}{n_p} + \frac{1}{n_u}\right)} \leq C_\delta\left(\frac{2\pi}{\sqrt{n_p}} + \frac{1}{\sqrt{n_u}}\right) = C_\delta\chi_{n_p,n_u}.$$

On the other hand, the deviation bound Eq.(15) is given due to

$$\left|\widehat{R}_{\text{DC-PU}}(g) - R(g)\right| \leq \left|\widehat{R}_{\text{DC-PU}}(g) - \widehat{R}_{uPU}(g)\right| + \left|\widehat{R}_{uPU}(g) - R(g)\right|,$$

where $\left|\widehat{R}_{\text{DC-PU}}(g) - \widehat{R}_{uPU}(g)\right| \geq 0$ with the probability at most $\Delta$, and $\left|\widehat{R}_{uPU}(g) - R(g)\right|$ shares the same concentration inequality with $\left|\widehat{R}_{\text{DC-PU}}(g) - \mathbb{E}[\widehat{R}_{\text{DC-PU}}(g)]\right|$. $\qquad\square$

## A.3 Proof of Theorem 3.3

*Proof.* We follow the spirit of proving Lemma 5 and Theorem 4 in [12] to give the proof of Theorem 3.3. Specifically, with the assumption of $\inf_{g \in \mathcal{G}} R_n^-(g) \geq \alpha > 0$, Theorem 3.2 and McDiarmid's inequality, we have

$$\sup_{g \in \mathcal{G}}\left|\widehat{R}_{\text{DC-PU}}(g) - R(g)\right| \leq \sup_{g \in \mathcal{G}}\left|\widehat{R}_{\text{DC-PU}}(g) - \mathbb{E}\left[\widehat{R}_{\text{DC-PU}}(g)\right]\right| + \pi' C_\ell \Delta, \qquad (18)$$

and with probability at least $1 - \delta$

$$\sup_{g \in \mathcal{G}}\left|\widehat{R}_{\text{DC-PU}}(g) - \mathbb{E}\left[\widehat{R}_{\text{DC-PU}}(g)\right]\right| - \mathbb{E}\left[\sup_{g \in \mathcal{G}}\left|\widehat{R}_{\text{DC-PU}}(g) - \mathbb{E}\left[\widehat{R}_{\text{DC-PU}}(g)\right]\right|\right] \leq C_\delta'\chi_{n_p,n_u}. \quad (19)$$

Given a ghost sample $(\mathcal{D}_p', \mathcal{D}_u')$, it holds

$$\mathbb{E}\left[\sup_{g \in \mathcal{G}}\left|\widehat{R}_{\text{DC-PU}}(g) - \mathbb{E}\left[\widehat{R}_{\text{DC-PU}}(g)\right]\right|\right] \leq \mathbb{E}_{(\mathcal{D}_p,\mathcal{D}_u),(\mathcal{D}_p',\mathcal{D}_u')}\left[\sup_{g \in \mathcal{G}}\left|\widehat{R}_{\text{DC-PU}}(g; \mathcal{D}_p, \mathcal{D}_u) - \widehat{R}_{\text{DC-PU}}(g; \mathcal{D}_p', \mathcal{D}_u')\right|\right].$$
$$(20)$$

The main difference is to decompose $\left|\widehat{R}_{\text{DC-PU}}(g; \mathcal{D}_p, \mathcal{D}_u) - \widehat{R}_{\text{DC-PU}}(g; \mathcal{D}_p', \mathcal{D}_u')\right|$. For the DC-PU risk given in Eq.(8), we have

$$\left|\widehat{R}_{\text{DC-PU}}(g; \mathcal{D}_p, \mathcal{D}_u) - \widehat{R}_{\text{DC-PU}}(g; \mathcal{D}_p', \mathcal{D}_u')\right|$$

$$= \left|\pi\widehat{R}_p^+(g; \mathcal{D}_p) - \pi\widehat{R}_p^+(g; \mathcal{D}_p') + \max\left\{\omega, \widehat{R}_u^-(g; \mathcal{D}_u) - \pi\widehat{R}_p^-(g; \mathcal{D}_p)\right\} - \max\left\{\omega', \widehat{R}_u^-(g; \mathcal{D}_u') - \pi\widehat{R}_p^-(g; \mathcal{D}_p')\right\}\right|$$

$$= \left|\pi\widehat{R}_p^+(g; \mathcal{D}_p) - \pi\widehat{R}_p^+(g; \mathcal{D}_p')\right.$$

$$\left. + \max\left\{0, \widehat{R}_u^-(g; \mathcal{D}_u) - \pi\widehat{R}_p^-(g; \mathcal{D}_p) - \omega\right\} - \max\left\{0, \widehat{R}_u^-(g; \mathcal{D}_u') - \pi\widehat{R}_p^-(g; \mathcal{D}_p') - \omega'\right\} + \omega - \omega'\right|$$

$$\leq \pi\left|\widehat{R}_p^+(g; \mathcal{D}_p) - \widehat{R}_p^+(g; \mathcal{D}_p')\right| + \pi\left|\widehat{R}_p^-(g; \mathcal{D}_p) - \widehat{R}_p^-(g; \mathcal{D}_p')\right| + \left|\widehat{R}_u^-(g; \mathcal{D}_u) - \widehat{R}_u^-(g; \mathcal{D}_u')\right|$$

$$+ 2(1 - \pi)\left|\widehat{R}_p^+(g; \mathcal{D}_p) - \widehat{R}_p^+(g; \mathcal{D}_p')\right|$$

$$= (2 - \pi)\left|\widehat{R}_p^+(g; \mathcal{D}_p) - \widehat{R}_p^+(g; \mathcal{D}_p')\right| + \pi\left|\widehat{R}_p^-(g; \mathcal{D}_p) - \widehat{R}_p^-(g; \mathcal{D}_p')\right| + \left|\widehat{R}_u^-(g; \mathcal{D}_u) - \widehat{R}_u^-(g; \mathcal{D}_u')\right|,$$

where we employed $|\max\{0, z\} - \max\{0, z'\}| \leq |z - z'|$ and the fact that $\omega$ is decided by $(1 - \pi)\widehat{R}_p^+(g; \mathcal{D}_p)$. This decomposition results in

$$
\mathbb{E}\left[\sup_{g \in \mathcal{G}}\left|\widehat{R}_{\text{Dc-PU}}(g) - \mathbb{E}\left[\widehat{R}_{\text{Dc-PU}}(g)\right]\right|\right] \leq (2 - \pi)\mathbb{E}_{\mathcal{D}_p, \mathcal{D}_p'}\left[\sup_{g \in \mathcal{G}}\left|\widehat{R}_p^+(g; \mathcal{D}_p) - \widehat{R}_p^+(g; \mathcal{D}_p')\right|\right]
$$
$$
+ \pi\mathbb{E}_{\mathcal{D}_p, \mathcal{D}_p'}\left[\sup_{g \in \mathcal{G}}\left|\widehat{R}_p^-(g; \mathcal{D}_p) - \widehat{R}_p^-(g; \mathcal{D}_p')\right|\right]
$$
$$
+ \mathbb{E}_{\mathcal{D}_u, \mathcal{D}_u'}\left[\sup_{g \in \mathcal{G}}\left|\widehat{R}_u^-(g; \mathcal{D}_u) - \widehat{R}_u^-(g; \mathcal{D}_u')\right|\right] \quad (21)
$$

Combining Eqs.(18), (19), (21) and the proof of Lemma 5 in [12], for any $\delta > 0$, it holds with probability at least $1 - \delta$

$$
\sup_{g \in \mathcal{G}}\left|\widehat{R}_{\text{Dc-PU}}(g) - R(g)\right| \leq 8L_\ell\mathfrak{R}_{n_p, p_p}(\mathcal{G}) + 4L_\ell\mathfrak{R}_{n_u, p}(\mathcal{G}) + C_\delta'\chi_{n_p, n_u} + \pi'C_\ell\Delta. \quad (22)
$$

Then we obtain the estimation bound through

$$
R(\widehat{g}_{\text{Dc-PU}}) - R(g^*) = \left(\widehat{R}_{\text{Dc-PU}}(\widehat{g}_{\text{Dc-PU}}) - \widehat{R}_{\text{Dc-PU}}(g^*)\right)
$$
$$
+ \left(R(\widehat{g}_{\text{Dc-PU}}) - \widehat{R}_{\text{Dc-PU}}(\widehat{g}_{\text{Dc-PU}})\right)
$$
$$
+ \left(\widehat{R}_{\text{Dc-PU}}(g^*) - R(g^*)\right)
$$
$$
\leq 0 + 2\sup_{g \in \mathcal{G}}\left|\widehat{R}_{\text{Dc-PU}}(g) - R(g)\right|
$$
$$
\leq 16L_\ell\mathfrak{R}_{n_p, p_p}(\mathcal{G}) + 8L_\ell\mathfrak{R}_{n_u, p}(\mathcal{G}) + 2C_\delta'\chi_{n_p, n_u} + 2\pi'C_\ell\Delta,
$$

where $\widehat{R}_{\text{Dc-PU}}(\widehat{g}_{\text{Dc-PU}}) \leq \widehat{R}_{\text{Dc-PU}}(g^*)$ by the definition of $\widehat{g}_{\text{Dc-PU}}$. $\qquad\square$

# B Baseline Methods

To evaluate the balance between positive and negative classification error rates of our proposed risk estimator, we select five classical risk estimator methods for comparison: uPU [11], nnPU [12], abs-PU [15], Dist-PU [13], and FOPU [18].

- **U**nbiased **P**ositive-**U**nlabeled Learning (**uPU**) [11]: An unbiased and convex optimization framework for PU learning, eliminating bias through the strategic application of distinct loss functions to positive and unlabeled samples.

$$
\widehat{R}_{\text{uPU}}(g) = \pi\widehat{R}_p^+(g) + \widehat{R}_u^-(g) - \pi\widehat{R}_p^-(g)
$$

- **P**ositive-**U**nlabeled Learning with **N**on-**N**egative Risk Estimator (**nnPU**) [12]: it introduces a non-negative risk estimator that addresses the overfitting problem in PU learning by explicitly constraining empirical risks to be non-negative.

$$
\widehat{R}_{\text{nnPU}}(g) = \pi\widehat{R}_p^+(g) + \max\{0, \widehat{R}_u^-(g) - \pi\widehat{R}_p^-(g)\}
$$

- **Abs**olute **P**ositive-**U**nlabeled Learning (**abs-PU**) [15]: it simplifies the non-negative risk constraint in nnPU through absolute-value correction.

$$
\widehat{R}_{\text{abs}-\text{PU}}(g) = \pi\widehat{R}_p^+(g) + \left|\widehat{R}_u^-(g) - \pi\widehat{R}_p^-(g)\right|
$$

- **P**ositive-**U**nlabeled Learning from a Label **Dist**ribution Perspective (**Dist-PU***) [13]: it learns a classifier by aligning the predicted and ground-truth label distributions, mitigating the negative-prediction bias prevalent in conventional PU learning methods.

$$
\widehat{R}_{\text{Dist}-\text{PU}}(g) = 2\pi\widehat{R}_p^+(g) + \left|\widehat{R}_u^-(g) - \pi\right|
$$

- **F**airness-Aware **O**nline **P**ositive-**U**nlabeled Learning (**FOPU**[*]) [18]: it addresses fairness issues in text classification by introducing a convex fairness constraint, improving prediction fairness across different demographic groups while maintaining model performance. Additionally, we adapt the fairness constraint mechanism from FOPU, originally designed for sensitive attributes, to the framework of class fairness.

$$\widehat{R}_{\text{FOPU}}(g) = \widehat{R}_{\text{nnPU}} + \widehat{R}_{\text{fair}}$$

Furthermore, to verify the effectiveness of DC-PU, we select three PU additional learning baselines, including MIXPUL [30], PULNS [7], and P$^3$Mix [6].

- **C**onsistency-based **A**ugmentation for **P**ositive and **U**nlabeled **L**earning (**MIXPUL**) [30]: it achieves data augmentation in positive-unlabeled learning by combining consistency regularization and reliable negative mining without requiring class prior probabilities. The code provided by its authors is utilized with default parameters, *i.e.* $\alpha = 1$, $\beta = 1$ and $\eta = 1$.
- **P**ositive-**U**nlabeled **L**earning with Effective **N**egative Sample **S**elector (**PULNS**) [7]: it optimizes the negative example selector through reinforcement learning to select negative examples from unlabeled data and alternates training with the classifier to effectively handle label noise issues. We take $\alpha = 1$ and $\beta = \{0.4, 0.6, 0.8, 1.0\}$ as suggested in its paper.
- **P**ositive and unlabeled learning with **P**artially **P**ositive **Mix**up (**P$^3$Mix**) [6]: it proposes a heuristic mixup technique that selects appropriate positive samples to mix with pseudo-negative samples near the classification boundary, achieving both data augmentation and supervision correction. This approach helps align the learning boundary closer to the supervised boundary. Additionally, two variants are introduced to enhance model robustness: **P$^3$Mix-E**, which employs a mean teacher to generate auxiliary target vectors for early-learning regularization, and **P$^3$Mix-C**, which directly corrects the labels of high-confidence pseudo-negative samples.
- **La**tent **G**roup-**A**ware **M**eta Disambiguation (**LaGAM**) [31]: it addresses PU learning by focusing on representation quality. It uses a hierarchical contrastive learning module to extract underlying group semantics from PU data, producing more discriminative representations. LaGAM then employs meta-learning to iteratively refine pseudo-labels of unlabeled data.

## C  Comparing with Practical PU learning Methods

To further validate the effectiveness of our proposed method, we incorporated the additional regularization terms inspired by Dist-PU [13] into the aforementioned five risk estimators for comparison as well as our risk estimator. First, to prevent trivial solutions where predictions cluster around ambiguous values, an entropy minimization term $L_{ent}$ is incorporated that encourages more confident predictions on unlabeled data:

$$L_{ent} = -\frac{1}{n_u} \sum_{i=1}^{n_u} [(1 - g(\mathbf{x}_i)) \log(1 - g(\mathbf{x}_i)) + g(\mathbf{x}_i) \log g(\mathbf{x}_i)] \tag{23}$$

Second, to address the critical issue of confirmation bias, where incorrect early predictions get reinforced during training, the mixup regularization term $L_{mix}$ is employed:

$$L_{mix} = \frac{1}{n} \sum_{i=1}^{n} \left[ \lambda' l_{\text{bce}}\big(g(\mathbf{x}_i'), g(\mathbf{x}_{1i})\big) + (1 - \lambda') l_{\text{bce}}\big(g(\mathbf{x}_i'), g(\mathbf{x}_{2i})\big) \right] \tag{24}$$

where mixed samples $\mathbf{x}_i' = \lambda' \mathbf{x}_{1i} + (1 - \lambda') \mathbf{x}_{2i}$ are created using mixing coefficients $\lambda$ sampled from a $\text{Beta}(\alpha, \alpha)$ distribution, with $\lambda' = \max(\lambda, 1 - \lambda)$. And $l_{bce}(t, t) = -(1 - t) \log(1 - t') - t \log(t')$. Additionally, an entropy minimization term $L_{ent}'$ is introduced to the mixed samples themselves:

$$L_{ent}' = \frac{1}{n} \sum_{i=1}^{n} l_{\text{bce}}\big(g(\mathbf{x}_i'), g(\mathbf{x}_i')\big) \tag{25}$$

Then, the additional regularization function is combined through hyperparameters $\mu_1$, $\mu_2$ and $\mu_3$ as:

$$L_{reg} = \mu_1 L_{ent} + \mu_2 L_{mix} + \mu_3 L_{ent}' \tag{26}$$

Table 5: Results of Micro-F1 (mean±std) on four benchmark PU datasets after adding the regularization term. The highest scores among PU learning methods are indicated in **bold**.

| Method | CIFAR-10-1 | CIFAR-10-2 | F-MNIST-1 | F-MNIST-2 |
|---|---|---|---|---|
| uPU[+] | 89.38±0.001 | 82.73±0.106 | 91.55±0.001 | 86.33±0.004 |
| nnPU[+] | 89.46±0.001 | 87.66±0.001 | 91.66±0.001 | 88.41±0.001 |
| abs-PU[+] | 89.45±0.001 | 88.09±0.000 | 91.68±0.001 | 88.57±0.009 |
| Dist-PU | 89.16±0.002 | 88.51±0.001 | 91.70±0.000 | 88.68±0.006 |
| FOPU[*+] | 89.51±0.002 | 87.30±0.000 | 91.65±0.001 | 88.39±0.009 |
| MIXPUL | 87.00±1.900 | 87.00±1.100 | 87.50±1.500 | 89.00±0.500 |
| PULNS | 87.20±0.600 | 83.70±2.900 | 90.70±0.500 | 87.90±0.500 |
| P$^3$Mix-E | 88.20±0.400 | 84.70±0.500 | 91.90±0.300 | **89.50±0.500** |
| P$^3$Mix-C | 88.70±0.400 | 87.90±0.500 | 92.00±0.400 | 89.40±0.300 |
| LaGAM | 89.91±0.300 | 87.98±1.400 | 90.15±0.013 | 80.88±0.084 |
| **DC-PU[+]** | **89.54±0.002** | **88.99±0.001** | **92.73±0.001** | 89.41±0.005 |

The experimental results are presented in Table 5. The methods marked with "+" indicate the incorporation of additional regularization terms. As shown in Table 5, DC-PU $^+$ achieves highly competitive performance across all four benchmark datasets. Notably, on the F-MNIST-1 dataset, DC-PU + reaches the highest Micro-F1 of 92.73%, significantly outperforming other methods. The only exception occurs on F-MNIST-2, where DC-PU $^+$ achieves a high Micro-F1 of 89.41% but slightly trails behind 89.50% of P$^3$Mix-E, with a marginal difference of only 0.09 percentage points. Moreover, all methods marked with "+" demonstrate more stable performance compared to their base versions. These results validate that incorporating the regularization term from Dist-PU can indeed enhance model performance, not only improving Micro-F1 but also strengthening model stability.

## D  Limitations and Broader Impacts

### D.1  Limitations

While DC-PU is evaluated on both synthetic and real-world datasets (*e.g.*, Alzheimer), most benchmark datasets used in our study (such as CIFAR-10 and F-MNIST) are originally designed for multi-class classification and are converted into binary classification tasks by partitioning class labels. This transformation simplifies the evaluation setting and may not fully reflect the complexity of real-world positive-unlabeled scenarios involving inherently multi-label or structured data. In future work, we plan to explore the application of DC-PU in more complex PU learning settings beyond binary classification.

Furthermore, our definition of fairness focuses solely on equalizing risks between positive and negative classes. This form of fairness does not incorporate demographic or group-level considerations, which are essential in socially sensitive applications. Extending DC-PU to address group fairness or fairness with respect to sensitive attributes remains an important future direction.

### D.2  Broader Impacts

The DC-PU method proposed in this paper addresses fairness issues in positive-unlabeled learning by mitigating the bias of traditional risk estimators toward negative classes. This method enhances the fairness and reliability of PU risk estimators. The method can be widely applied in practical applications (such as medical screening, recommendation systems, fraud detection, etc.), improving model stability and reliability in real-world scenarios where negative samples are difficult to obtain. However, it's important to acknowledge that the DC-PU method relies on specific sampling assumptions (*i.e.*, SCAR assumption and one-sample assumption), which may not fully hold in some real-world applications.

