# OpenReview forum: "Balancing Positive and Negative Classification Error Rates in Positive-Unlabeled Learning"
_NeurIPS.cc/2025/Conference — NeurIPS 2025 poster_

### Official Review · Reviewer_k4vm · 2025-06-21

**Clarity:** 2
**Significance:** 3
**Originality:** 3
**Rating:** 4
**Confidence:** 3

**Summary:**

This paper studies the Positive-Unlabeled (PU) learning with a dual-constrained risk estimator. Their approach is named by DC-PU. The idea stems from a question about imbalance classification error rates in nnPU. Then, they try to balance the risks of positive-class and negative-class and the theoretical analysis is presented. They train DC-PU on the augmented Lagrange multiplier framework and evaluate it with the Fashion-MNIST and CIFAR-10 datasets.

**Questions:**

I wonder how their approach applies to the recent dataset DIABETES from [J. Gardner, Z. Popovic, and L. Schmidt. Benchmarking distribution shift in tabular data with tableshift. (NeurIPS 2023)].

**Ethical Concerns:**

["NO or VERY MINOR ethics concerns only"]

**Final Justification:**

I thank the authors' efforts.

They presented detailed response to my questions. In particular, I appreciated the performance of their method DC-PU on DIABETES.

I therefore increase the score and the confidence.

**Limitations:**

yes

**Quality:**

3

**Strengths And Weaknesses:**

Strengths:
1. The authors propose a new approach called DC-PU for PU learning. DC-PU is a new risk estimator for PU learning that explicitly addresses the imbalance in classification error rates between positive and negative classes.
2. The proposed method is motivated by empricial observations of imbalanced error rates in nnPU. This gives a practical solution to real-world applications where such imbalances occur.

Weaknesses:
1. While the theoretical analysis is rigorous, the paper does not clearly explain how these theoretical results translate into the improved performance observed in experiments. For instance, how do the derived bounds in Section 3 directly influence the training dynamics or generalization of DC-PU? A deeper discussion linking theory to practice would strengthen the paper. I suspect that the theoretical analysis is not necessary to state their main claim.
2. The ablation study (in Section 5.4) evaluates the impact of two constraints (weak and strong). However, they did not mention the choice of other hyperparameters (such as β in Eq. (10)).

---

> ### Author Rebuttal · Authors · 2025-07-31
>
> **Q1. How these theoretical results translate into the improved performance observed in experiments?**
>
> Thank you for your suggestion. Our theoretical framework provides direct mathematical explanations for the experimental phenomena.
>
> **The training stability**. Our bias analysis in Lemma 3.1 reveals that DC-PU's positive bias occurs with exponentially decaying probability, which directly explains the improved training stability observed in our experiments. While uPU suffers from severe overfitting because $\widehat{R}\_u^-(g)-\pi \widehat{R}\_p^-(g)$ tends to be less than 0 during model training, and non-negativity constraint of nnPU leads to the negative-class risk estimator frequently approaching zero, DC-PU's dual-constraint mechanism prevents both problems with high probability. The asymptotic convergence properties further illuminate this advantage that the generalization error bound in Theorem 3.3 contains four major terms, including two Rademacher complexity terms $\mathfrak{R}\_{n_u,p}(\mathcal{G})$ and $\mathfrak{R}\_{n_p,p_p}(\mathcal{G})$, a sampling fluctuation term $\chi_{n_p,n_u}$, and a distribution discrepancy term $\Delta$. Under standard assumptions, as $n_p, n_u \to \infty$, the Rademacher complexity terms and sampling fluctuation term converge to zero while the bias term $\Delta$ remains constant, consequently ensuring $R(\widehat{g}_{\mathrm{DC-PU}}) \to R(g^*)$. The convergence rates analysis shows that the overall rate is characterized by $O(\max(1/\sqrt{n_p}, 1/\sqrt{n_u}))$, which explains the smooth and consistent learning curves we observe in Figures 3 and 4.
>
> **The generalization**. The estimation error bound in Theorem 3.3 provides explicit guidance on how our design choices improve generalization performance. The controlled bias term ensures that DC-PU doesn't suffer from the systematic over-play of negative-class risk. This theoretical framework explains why DC-PU consistently achieves superior "last" performance metrics in Tables 2 and 3. As the exponentially small bias probability ensures robust behavior regardless of specific dataset characteristics.
>
> **The GAP**. The explicit equality constraint $\hat{R}_p^+(g) = \frac{\hat{R}_u^-(g) - \pi\hat{R}_p^-(g)}{1-\pi}$ in our DC-PU theoretically guarantees that the empirical risks of positive and negative classes remain close throughout training. This mathematical constraint directly translates into the consistently low GAP values shown in Figure 2, where DC-PU maintains balanced error rates across all datasets.
>
>
> &nbsp;
>
>
> **Q2. Lack the choice of other hyperparameters.**
>
> Thank you for your comment. We conduct comprehensive parameter sensitivity analysis. For the parameter $\tau$, our experiments show optimal performance at $2\times10^{-3}$ across most datasets, which aligns with our theoretical analysis that moderate penalty parameters achieve balance between constraint strength and optimization stability. For the parameter $\beta$, we find that the range $[0.4,0.5]$ performs optimally across different datasets, which is significant because $\beta$ controls the update speed of dynamic lower bound. Too small value of $\beta$ leads to overly loose constraints losing the balancing effect, while too large value of $\beta$ causes optimization instability. Our experiments demonstrate that DC-PU exhibits relative robustness to these hyperparameters, with parameter variations within reasonable ranges not significantly affecting performance, further validating the practicality of our method.
>
> | $\tau$ | cifar10-1 | cifar10-2 | fmnist-1 | fmnist-2 |
> |:---------:|:---------:|:---------:|:--------:|:--------:|
> | $2\times10^{-3}$ | **89.37** | **88.6** | 91.21 | 88.45 |
> | $2\times10^{-2}$ | 89.31 | 87.8 | 91.23 | 88.06 |
> | $2\times10^{-1}$ | 89.18 | 88.06 | **91.48** | **88.91** |
> | 0 | 88.54 | 88.36 | 91.28 | 86.97 |
> | $2\times10^{0}$ | 88.94 | 86.52 | 91.32 | 86.44 |
> | $2\times10^{1}$ | 88.53 | 69.98 | 90.89 | 66.31 |
> | $2\times10^{2}$ | 67.82 | 65.07 | 85.3 | 69.61 |
> | $2\times10^{3}$ | 65.75 | 60.67 | 59.8 | 65.27 |
>
> &nbsp;
>
> | $\beta$ | cifar10-1 | cifar10-2 | fmnist-1 | fmnist-2 |
> |:-----:|:---------:|:---------:|:--------:|:--------:|
> | 0 | 88.62 | 87.59 | 91.18 | 86.92 |
> | 0.1 | 87.91 | 87.8 | 91.38 | 87.37 |
> | 0.2 | 88.33 | 88.07 | 91.24 | 87.41 |
> | 0.3 | 88.75 | 87.97 | 91.35 | 87.8 |
> | 0.4 | 88.49 | 88.51 | **91.48** | **88.91** |
> | 0.5 | **89.37** | **88.6** | 91.27 | 88.13 |
> | 0.6 | 89.2 | 87.9 | 91.23 | 87.77 |
> | 0.7 | 88.82 | 87.98 | 91.22 | 87.84 |
> | 0.8 | 88.62 | 87.82 | 91.26 | 87.38 |
> | 0.9 | 88.63 | 87.68 | 91.28 | 87.22 |
> | 1 | 88.47 | 86.49 | 91.4 | 87.5 |
>
> &nbsp;
> &nbsp;
>
>
> **Q3. How DC-PU applies to the recent dataset DIABETES?**
>
> Thank you for your comment. We add the experimental results on the DIABETES dataset, as shown below. The results on DIABETES dataset provide compelling evidence for the effectiveness of our method in real-world tabular data scenarios. It demonstrates consistent performance throughout training and indicates that DC-PU maintains balanced performance across both classes, which is crucial for real-world medical applications. The most significant finding lies in the GAP metrics: 4.78±0.009 (min) and 10.12±0.009 (last), substantially lower than competing methods like uPU and nnPU. This dramatic reduction in classification error imbalance demonstrates that our dual-constraint mechanism effectively addresses the fundamental challenge of imbalanced error rates even in complex tabular data with intricate feature interactions. The consistent low GAP values across all metrics confirm that our mathematical constraints translate effectively to practical performance improvements in scenarios.
>
> | Method     | Micro-F1 (max) | Micro-F1 (last) | Macro-F1 (max) | Macro-F1 (last) | GAP (min)  | GAP (last)  |
> |------------|----------------|-----------------|----------------|-----------------|------------|-------------|
> | uPU        | 59.09±0.004    | **57.59±0.001** | 51.23±0.028    | 38.37±0.006     | 10.46±0.017 | 40.78±0.006 |
> | nnPU       | 59.08±0.004    | 55.41±0.006     | 55.82±0.006    | 54.33±0.006     | 5.71±0.011  | 10.76±0.008 |
> | abs-PU     | 59.08±0.004    | 55.61±0.004     | 55.70±0.009    | 54.39±0.006     | 5.95±0.013  | 11.39±0.012 |
> | Dist-PU*   | 58.34±0.002    | 55.33±0.004     | 55.80±0.004    | 54.15±0.005     | 5.22±0.007  | 10.69±0.010 |
> | FOPU*      | **59.14±0.004**| 55.22±0.003     | 54.86±0.009    | 53.08±0.005     | 7.18±0.007  | 11.94±0.008 |
> | DC-PU      | 59.10±0.004    | 56.17±0.004     | **56.16±0.008**| **54.79±0.013** | **4.78±0.009** | **10.12±0.009** |

---

> > ### Comment · Reviewer_k4vm · 2025-08-06
> >
> > I thank the authors' efforts.
> >
> > They presented detailed response to my questions. In particular, I appreciated the performance of their method DC-PU on DIABETES.
> >
> > I therefore increase the score and the confidence.

---

> > > ### Author Response · Authors · 2025-08-06
> > >
> > > We are deeply grateful for your careful evaluation and for increasing your score. Your constructive feedback has been invaluable to improving our work.

---

### Official Review · Reviewer_cuct · 2025-06-22

**Clarity:** 3
**Significance:** 3
**Originality:** 3
**Rating:** 5
**Confidence:** 4

**Summary:**

This paper addresses the problem of imbalanced classification error rates in positive-unlabeled (PU) learning caused by the fact that the empirical risk estimates of the negative class of existing risk assessors (such as nnPU) approach 0. To solve this problem, the paper proposes a new risk assessor called DC-PU, which aims to balance the classification error rates of positive and negative classes. This method enforces the empirical risk estimate of the negative class to be close to the empirical risk of the positive class through a dynamic lower bound and an explicit equality constraint. Experiments have shown that DC-PU can achieve higher accuracy and more stable convergence than other risk assessors.

**Questions:**

1. Given that the strong constraint enforces strict equality between positive and negative class risks, is the weak constraint still necessary? Could you clarify the individual contributions and potential redundancy between these two constraints?
2. Theorems 3.2 and 3.3 present conclusions without adequate analysis. Could you provide more insight into aspects such as convergence rate analysis of the established bounds?
3. On certain datasets, your method shows inferior "max" performance compared to baseline methods, while achieving superior "last" performance. How do you interpret this phenomenon?

**Ethical Concerns:**

["NO or VERY MINOR ethics concerns only"]

**Final Justification:**

Consider the response from authors and rebuttal from other reviewers, I keep my original score.

**Limitations:**

yes

**Quality:**

3

**Strengths And Weaknesses:**

Strengths:
1. The authors provide a clear empirical demonstration of the imbalance problem between FPR and FNR in nnPU through comprehensive experiments, which effectively motivates their proposed solution with dual constraints.
2. The authors establish solid theoretical guarantees including bias analysis (Lemma 3.1), consistency analysis (Theorem 3.2), and estimation error bounds (Theorem 3.3), providing a comprehensive theoretical understanding of the proposed method.
3. Beyond maintaining traditional performance metrics, the authors introduce a novel GAP metric to specifically measure the balance between FPR and FNR.

Weaknesses:
1. The assumptation that the expected risks of the positive-class and negative-class should be close may not be true in some situations. I think this fundamental assumption is overly idealistic. So the authors should further emphasizing the limitations of this aspect.
2. Lack of sensitivity analysis of introduced parameters, such as $\tau$ and $\beta$.
3. While detailed proofs are provided in the appendix, the main text lacks sufficient analysis and interpretation of the theoretical results. The authors should provide more intuitive explanations and summarizing conclusions regarding the implications of their theoretical findings.

---

> ### Author Rebuttal · Authors · 2025-07-31
>
> **Q1. Lack of sensitivity analysis of introduced parameters.**
>
> Thank you for your comment. We conduct comprehensive parameter sensitivity analysis. For the parameter $\tau$, our experiments show optimal performance at $2\times10^{-3}$ across most datasets, which aligns with our theoretical analysis that moderate penalty parameters achieve balance between constraint strength and optimization stability. For the parameter $\beta$, we find that the range $[0.4,0.5]$ performs optimally across different datasets, which is significant because $\beta$ controls the update speed of dynamic lower bound. Too small value of $\beta$ leads to overly loose constraints losing the balancing effect, while too large value of $\beta$ causes optimization instability. Our experiments demonstrate that DC-PU exhibits relative robustness to these hyperparameters, with parameter variations within reasonable ranges not significantly affecting performance, further validating the practicality of our method.
>
> &nbsp;
>
> | $\tau$ | cifar10-1 | cifar10-2 | fmnist-1 | fmnist-2 |
> |:---------:|:---------:|:---------:|:--------:|:--------:|
> | $2\times10^{-3}$ | **89.37** | **88.6** | 91.21 | 88.45 |
> | $2\times10^{-2}$ | 89.31 | 87.8 | 91.23 | 88.06 |
> | $2\times10^{-1}$ | 89.18 | 88.06 | **91.48** | **88.91** |
> | 0 | 88.54 | 88.36 | 91.28 | 86.97 |
> | $2\times10^{0}$ | 88.94 | 86.52 | 91.32 | 86.44 |
> | $2\times10^{1}$ | 88.53 | 69.98 | 90.89 | 66.31 |
> | $2\times10^{2}$ | 67.82 | 65.07 | 85.3 | 69.61 |
> | $2\times10^{3}$ | 65.75 | 60.67 | 59.8 | 65.27 |
>
> &nbsp;
>
> | $\beta$ | cifar10-1 | cifar10-2 | fmnist-1 | fmnist-2 |
> |:-----:|:---------:|:---------:|:--------:|:--------:|
> | 0 | 88.62 | 87.59 | 91.18 | 86.92 |
> | 0.1 | 87.91 | 87.8 | 91.38 | 87.37 |
> | 0.2 | 88.33 | 88.07 | 91.24 | 87.41 |
> | 0.3 | 88.75 | 87.97 | 91.35 | 87.8 |
> | 0.4 | 88.49 | 88.51 | **91.48** | **88.91** |
> | 0.5 | **89.37** | **88.6** | 91.27 | 88.13 |
> | 0.6 | 89.2 | 87.9 | 91.23 | 87.77 |
> | 0.7 | 88.82 | 87.98 | 91.22 | 87.84 |
> | 0.8 | 88.62 | 87.82 | 91.26 | 87.38 |
> | 0.9 | 88.63 | 87.68 | 91.28 | 87.22 |
> | 1 | 88.47 | 86.49 | 91.4 | 87.5 |
>
> &nbsp;
> &nbsp;
>
> **Q2. Lack sufficient analysis of the theoretical results.**
>
> Thank you for your valuable suggestion. Theorem 3.3 establishes the fundamental generalization bound that explains DC-PU's superior performance through four distinct components: two Rademacher complexity terms $\mathfrak{R}\_{n_u,p}(\mathcal{G})$ and $\mathfrak{R}\_{n_p,p_p}(\mathcal{G})$, a sampling fluctuation term $\chi_{n_p,n_u}$, and a distribution discrepancy term $\Delta$.
>
> **The asymptotic convergence properties.** Under standard assumptions, as $ n_p, n_u \to \infty$, two Rademacher complexity terms gradually decrease and converge to zero, and so does the sampling fluctuation term $\chi_{n_p,n_u}$. The bias term $\Delta$ is independent of sample size and can be treated as a constant. Consequently, as $n\to\infty$, $R(\widehat{g}_{\mathrm{DC-PU}}) \to R(g^*)$.
>
> **The convergence rates.** The convergence rates of the first and second terms are governed by the Rademacher complexities $\mathfrak{R}\_{n_u,p}(\mathcal{G})$ and $\mathfrak{R}\_{n_p,p_p}(\mathcal{G})$, corresponding to rates of $O(1/\sqrt{n_p})$ and $O(1/\sqrt{n_u})$, respectively. The fluctuation term $\chi_{n_p,n_u}$ contributes an additional rate of $O(1/\sqrt{n_p}+1/\sqrt{n_u})$. Consequently, the overall convergence rate is characterized by $ O\left(\max(1/\sqrt{n_p},1/\sqrt{n_u})\right)$.  It indicates that DC-PU achieves minimax optimal convergence while maintaining balanced performance, contrasting with methods that optimize overall accuracy at the expense of class balance.
>
> Additionally, the exponential decay rate of bias probability provides crucial insight into DC-PU's stability advantages. It shows that when classes have natural difficulty separation $\alpha-\beta > 0$, our constraint mechanism provides exponentially strong concentration guarantees. This explains why DC-PU performs particularly well on datasets with inherent class differences.
>
> **The sampling fluctuation term.** The sampling fluctuation term $\chi_{n_p,n_u} = 2\pi/\sqrt{n_p} + 1/\sqrt{n_u}$ provides explicit mathematical guidance for experimental design. This term shows that when positive class prior $\pi$ is small, positive sample size $n_p$ becomes more critical for maintaining tight bounds, while unlabeled sample size $n_u$ provides consistent benefit regardless of class imbalance.
>
> **The distribution discrepancy term.** The distribution discrepancy term $\Delta$ represents the controlled bias introduced by our constraint mechanism. Unlike uncontrolled bias in traditional methods, this term decreases exponentially with probability, ensuring that our method maintains consistency while providing stability benefits. It explicitly connects sample sizes, class difficulty separation, and theoretical guarantees:
> $$\Delta \leq \pi' C_\ell \exp(-2(1-\pi)^2(\alpha-\beta)^2/C_\ell^2 \cdot 1/(\pi^2/n_p + 1/n_u))$$
>
>
> &nbsp;
>
>
> **Q3. Is the weak constraint still necessary?**
>
> Thank you for your comments. We argue that the weak constraint cannot be considered as being incorporated within the strong constraint.
>
> First, since the expected risk is generally intractable, empirical risk is commonly used as its approximation. Compared to the strong constraint, which enforces strict equality between the empirical risks of the positive and negative classes, and thereby introduces potential estimation error. The weak constraint imposes only a lower bound on the negative class risk, which makes it a more tolerant and robust form of estimation under approximation.
>
> Second, from a convergence perspective, using the strong constraint alone tends to lead to rapid convergence and overfitting in the early stages of training. In contrast, the weak constraint contributes to a more stable training process and helps mitigate overfitting by introducing a lower-bound formulation.
>
> Finally, ablation studies indicate that the weak constraint can sometimes achieve better performance than the strong constraint, indicating that the two constraints are complementary in practice.
>
>
> &nbsp;
>
>
> **Q4. Why the method outperforms baselines in "last" but not in "max" on certain datasets?**
>
> Thank you for the suggestion. This phenomenon actually demonstrates the strength of our method rather than a limitation, as the "max" performance represents peak instantaneous performance that may include overfitting to particular batches, while "last" performance reflects stable, converged performance under full constraint enforcement that is more representative of true generalization capability. For example, the results of FOPU* on Alzheimer show that it tends to overfit in later stages while it can find a strong model during training, resulting in lower average performance in the final epochs. This indicates that the model trained by FOPU* is less stable than ours.

---

> ### Comment · Reviewer_cuct · 2025-08-06
>
> Thanks for the author's reply, I tend to keep the original score.

---

> > ### Author Response · Authors · 2025-08-06
> >
> > We are deeply grateful for your careful evaluation and for giving a positive score. Your constructive feedback has been invaluable to improving our work.

---

### Official Review · Reviewer_YFBq · 2025-06-28

**Clarity:** 2
**Significance:** 3
**Originality:** 3
**Rating:** 5
**Confidence:** 4

**Summary:**

This paper focuses on positive-unlabeled (PU) learning, an important problem in weakly supervised learning. It observes that the loss term with respect to the negative class is overly minimized, causing imbalanced learning of the positive and negative classes. To address this issue, the loss term is modified by raising the minimum value for positive data loss. Extensive theoretical analysis and experimental results validate the effectiveness of the proposed method.

**Questions:**

Please see "Weaknesses". My major question is regarding the main theorem.

**Ethical Concerns:**

["NO or VERY MINOR ethics concerns only"]

**Final Justification:**

Thanks for the rebuttal. Since the error was fixed during the rebuttal, the theoretical analysis looks good to me now. Therefore, I have increase my score. Please include the replies in the rebuttal to the final version of the paper.

**Limitations:**

The limitations of the paper should be further discussed.

**Paper Formatting Concerns:**

None.

**Quality:**

2

**Strengths And Weaknesses:**

## Strengths
- The studied problem is important to the literature.

- The proposed technique is simple yet effective.

- The theoretical analysis is comprehensive and novel, such as the consistency analysis.

- Extensive experiments validate the effectiveness of the proposed method.

## Weaknesses
- I had a question regarding the proof of Lemma 3.1, which shows that the bias will converge to zero as $n$ goes to infinity. In the last inequality in line 339, it seems that
\begin{equation}
(1-\pi)\alpha-(1-\pi)\hat{R}_p^+(g)\leq (1-\pi)(\alpha-\beta).
\end{equation}
Therefore, it seems that the $\leq$ should be $\geq$ and the inequality does not hold. It is a very important part since the consistency may not hold.


- The proposed technique is related to flooding, but the paper lacks discussion of this relationship. In flooding, a constant value is considered the minimum value of the risk-correction function. This paper proposes using a term proportional to the loss term of the positive class as the minimum value of the risk correction function, which could solve the hyperparameter selection problem. Further discussion would benefit the paper.

- The writing of the paper can be improved; for example, there are several typos. Additionally, the references can be revised for concision. Some minor points: 1) In line 13, it should read "we." 2) The notation "~" in line 110 is inaccurate. (3) In line 146, the infinity notation needs to be revised.

- Theorem 3.3 lacks discussion.

Reference:

[1] Do We Need Zero Training Loss After Achieving Zero Training Error?, ICML 2020.

---

> ### Author Rebuttal · Authors · 2025-07-31
>
> **Q1. Correction of the proof of Lemma 3.1.**
>
> Thank you for pointing out this error. It begins with a mistake in which we wrote the upper bound condition of $\widehat{R}\_p^+(g)$, i.e., $0 \le \sup_{g \in \mathcal{G}} \widehat{R}\_p^+(g) \le \beta $, as a lower bound condition in the current manuscript. When applying the correct condition $0 \le \sup_{g \in \mathcal{G}} \widehat{R}\_p^+(g) \le \beta$, the inequality $(1-\pi)R\_n^-(g)-\bigl(\widehat{R}\_u^-(g) - \pi \widehat{R}\_p^-(g)\bigr)\ge (1-\pi)(\alpha-\beta)$ is valid.
>
> This correction actually strengthens our theoretical analysis by demonstrating that the separation between the true negative class risk and our estimator depends on the gap $\alpha-\beta$ between the lower bounds of negative and positive class risks. The lemma shows that our bias probability bound $P(D_\omega^-(g)) \leq \exp(-2(1-\pi)^2(\alpha-\beta)^2/C_\ell^2 \cdot 1/(\pi^2/n_p + 1/n_u))$ has an exponential decay rate that explicitly depends on this separation. When $\alpha > \beta$, the exponential decay becomes stronger, providing better theoretical guarantees. It explains why DC-PU performs particularly well on datasets with natural class difficulty differences, as our constraint mechanism automatically adapts to these inherent class characteristics while maintaining theoretical consistency.
>
>
> &nbsp;
>
>
> **Q2. The paper lacks discussion of the relationship between flooding.**
>
> Thank you for your suggestion. Flooding uses a fixed constant flood level $\lambda$ where the loss becomes $\max\\{\ell(f(x), y)-\lambda,0\\}$, creating a uniform lower bound across all samples and classes. It addresses general overfitting by preventing zero loss but lacks class-specific considerations essential for PU learning scenarios. In contrast, our DC-PU employs a dynamic lower bound, creating a class-aware constraint that automatically adapts to the actual positive class risk. The exponential moving average ensures stability while allowing adaptation, contrasting sharply with flooding's static threshold.
>
> More critically, our explicit equality constraint $\widehat{R}_p^+(g) = \frac{\widehat{R}_u^-(g) - \pi \widehat{R}_p^-(g)}{1-\pi}$ introduces a mathematical relationship between classes that flooding cannot capture.
>
> Overall, DC-PU ensures that class-level risks remain proportionally balanced, addressing the fundamental challenge of PU learning where class imbalance emerges from the absence of labeled negative samples rather than general training instability.
>
>
> &nbsp;
>
>
> **Q3. There are several typos.**
>
> Thank you for your corrections. We will revise them in the next version.
>
>
> &nbsp;
>
>
> **Q4. Theorem 3.3 lacks discussion.**
>
> Thank you for your comment. Theorem 3.3 establishes the fundamental generalization bound that explains DC-PU's superior performance through four distinct components: two Rademacher complexity terms $\mathfrak{R}\_{n_u,p}(\mathcal{G})$ and $\mathfrak{R}\_{n_p,p_p}(\mathcal{G})$, a sampling fluctuation term $\chi_{n_p,n_u}$, and a distribution discrepancy term $\Delta$.
>
> **The asymptotic convergence properties.** Under standard assumptions, as $ n_p, n_u \to \infty$, two Rademacher complexity terms gradually decrease and converge to zero, and so does the sampling fluctuation term $\chi_{n_p,n_u}$. The bias term $\Delta$ is independent of sample size and can be treated as a constant. Consequently, as $n\to\infty$, $R(\widehat{g}_{\mathrm{DC-PU}}) \to R(g^*)$.
>
> **The convergence rates.** The convergence rates of the first and second terms are governed by the Rademacher complexities $\mathfrak{R}\_{n_u,p}(\mathcal{G})$ and $\mathfrak{R}\_{n_p,p_p}(\mathcal{G})$, corresponding to rates of $O(1/\sqrt{n_p})$ and $O(1/\sqrt{n_u})$, respectively. The fluctuation term $\chi_{n_p,n_u}$ contributes an additional rate of $O(1/\sqrt{n_p}+1/\sqrt{n_u})$. Consequently, the overall convergence rate is characterized by $ O\left(\max(1/\sqrt{n_p},1/\sqrt{n_u})\right)$.  It indicates that DC-PU achieves minimax optimal convergence while maintaining balanced performance, contrasting with methods that optimize overall accuracy at the expense of class balance.
>
> Additionally, the exponential decay rate of bias probability provides crucial insight into DC-PU's stability advantages. It shows that when classes have natural difficulty separation $\alpha-\beta > 0$, our constraint mechanism provides exponentially strong concentration guarantees. This explains why DC-PU performs particularly well on datasets with inherent class differences.
>
> **The sampling fluctuation term.** The sampling fluctuation term $\chi_{n_p,n_u} = 2\pi/\sqrt{n_p} + 1/\sqrt{n_u}$ provides explicit mathematical guidance for experimental design. This term shows that when positive class prior $\pi$ is small, positive sample size $n_p$ becomes more critical for maintaining tight bounds, while unlabeled sample size $n_u$ provides consistent benefit regardless of class imbalance.
>
> **The distribution discrepancy term.** The distribution discrepancy term $\Delta$ represents the controlled bias introduced by our constraint mechanism. Unlike uncontrolled bias in traditional methods, this term decreases exponentially with probability, ensuring that our method maintains consistency while providing stability benefits. It explicitly connects sample sizes, class difficulty separation, and theoretical guarantees:
> $$\Delta \leq \pi' C_\ell \exp(-2(1-\pi)^2(\alpha-\beta)^2/C_\ell^2 \cdot 1/(\pi^2/n_p + 1/n_u))$$

---

> > ### Comment · Reviewer_YFBq · 2025-08-05
> >
> > Thanks for the rebuttal. Since the error was fixed during the rebuttal, the theoretical analysis looks good to me now. Therefore, I have increase my score. Please include the replies in the rebuttal to the final version of the paper.

---

> > > ### Author Response · Authors · 2025-08-06
> > >
> > > We are deeply grateful for your careful evaluation again and for increasing your score. Your constructive feedback has been invaluable to improving our manuscript and theoretical analysis.

---

### Official Review · Reviewer_kGXk · 2025-07-02

**Clarity:** 3
**Significance:** 3
**Originality:** 3
**Rating:** 4
**Confidence:** 4

**Summary:**

This paper addresses an important and specific problem in Positive-Unlabeled (PU) learning: the imbalanced classification error rates produced by existing risk estimators, particularly the widely-used nnPU framework. The authors empirically demonstrate that classifiers trained with nnPU tend to have a False Positive Rate (FPR) that is significantly lower than the False Negative Rate (FNR), suggesting an over-emphasis on the negative class.
To tackle this issue, the paper proposes a novel risk estimator named DC-PU (Dual-Constrained PU). Its core idea is to enforce balance via a dual-constraint mechanism: 1) a "weak constraint" that sets a dynamic lower bound for the negative-class empirical risk, tying it to the positive-class empirical risk rather than a fixed value of 0; and 2) a "strong constraint" that introduces an explicit equality between the estimated positive- and negative-class risks. To optimize this constrained objective, the authors employ an augmented Lagrange multiplier (ALM) framework.

**Questions:**

Please correct the legends in Figures 2, 3, and 4, changing the incorrect "Fair-PU" label to "DC-PU."

**Ethical Concerns:**

["NO or VERY MINOR ethics concerns only"]

**Final Justification:**

The authors have successfully adressed the majority of my concerns as mentioned in the rebuttal.

**Limitations:**

1. Like much of the work in this area, the proposed method relies on the 'selected completely at random' (SCAR) assumption. It would be beneficial to briefly acknowledge in the main body of the paper that this assumption may not hold in all real-world scenarios, thus defining a boundary condition for the method's applicability.
2. The experiments are conducted on multi-class datasets that have been converted into binary PU tasks. While this is standard practice, these proxy datasets may not fully capture the complexities of native PU problems found in applications like information retrieval or medical diagnosis.

**Quality:**

3

**Strengths And Weaknesses:**

Strengths:
1. The research is well-targeted, addressing a specific and empirically demonstrated flaw in a strong baseline (nnPU). By first proving the existence of the error rate imbalance, the authors provide a compelling motivation for their work.
2. The core contribution, the dual-constraint mechanism in DC-PU, is an innovative and direct approach to solving the identified problem. Combining a dynamic lower bound with an explicit equality constraint is an elegant way to enforce balance at the objective function level.
3.  The experiments show that DC-PU not only excels at its primary goal of balancing error rates (GAP) but also achieves state-of-the-art or competitive performance in standard metrics like F1-scores and stability. This demonstrates a clear and multi-faceted improvement over existing methods.

Weaknesses:
1. The theoretical exposition is insufficient. The paper is formula-heavy, but the lack of detailed descriptions for these equations makes the content difficult to follow.
2. The legends in the main experimental figures (Figs 2, 3, 4) incorrectly label the proposed method as "Fair-PU" instead of "DC-PU." This is a critical error that is highly confusing for the reader.

---

> ### Author Rebuttal · Authors · 2025-07-31
>
> **Q1. The theoretical exposition is insufficient.**
>
> Thank you for your valuable suggestion. Our theoretical framework addresses a fundamental challenge in PU learning where existing methods suffer from the imbalanced error rates between positive and negative classes. The risk decomposition in Eq.3 reveals the core mathematical insight: under the SCAR assumption, we can express the negative class risk as $R_n^-(g) = \frac{R_u^-(g) - \pi R_p^-(g)}{1-\pi}$, which allows us to estimate negative class performance without labeled negative samples. However, this unbiased estimator $\widehat{R}_u^-(g) - \pi \widehat{R}_p^-(g) $ frequently becomes negative during training, leading existing methods like nnPU to truncate it at zero, resulting in the negative class being over-played.
>
> Our dual-constraint mechanism in Eq.8 fundamentally transforms this problem through two innovations. The dynamic lower bound $\omega = (1-\pi)\widehat{R}_p^+(g) $ automatically adapts to the current positive class performance, creating a balanced constraint that evolves with the model's learning progress. The explicit equality constraint $\widehat{R}_p^+(g) = \frac{\widehat{R}_u^-(g) - \pi \widehat{R}_p^-(g)}{1-\pi} $ mathematically enforces that the empirical positive and negative class risks remain equivalent throughout training, directly addressing the class imbalance problem at its theoretical root.
> The mathematical elegance lies in how these constraints work together: while the dynamic bound prevents degenerate solutions, the equality constraint ensures that neither class dominates the optimization landscape. This theoretical foundation explains why DC-PU achieves consistently low GAP values across most experimental settings, as the constraints translate directly into balanced error rates between positive and negative classes.
>
>
> &nbsp;
>
>
> **Q2. The legends are incorrectly labeled.**
>
> We sincerely apologize for this critical labeling error in figures. We have corrected all figure legends to properly reflect "DC-PU" as our method name and will ensure this correction is implemented in the next version.
>
>
> &nbsp;
>
>
> **Q3. The proposed method relies on the SCAR assumption, defining a boundary condition for the method's applicability.**
>
> Thank you for your suggestion. This is indeed a fundamental limitation that we should acknowledge more prominently in our main text. As the SCAR assumption can indeed be violated in real-world scenarios, we still believe our method provides robustness beyond strict SCAR assumption.
>
> Firstly, the dual-constraint mechanism inherently provides stability against moderate SCAR violations by preventing extreme class imbalance that could arise from distribution shift, while our dynamic lower bound adapts to the actual observed data distribution rather than relying solely on theoretical distributional assumptions.
>
> Furthermore, our evaluation on the real-world Alzheimer dataset represents a realistic medical diagnosis scenario where SCAR assumptions are likely violated to some degree, yet DC-PU still demonstrates clear improvements in balanced error rates and overall performance. We will expand our limitations discussion in the next version to explicitly address scenarios where SCAR may not hold.
>
>
> &nbsp;
>
>
> **Q4. The experiments are conducted on multi-class datasets that have been converted into binary PU tasks.**
>
> Thank you for your comment. In fact, in addition to the artificially converted datasets, our evaluation also includes the real-world Alzheimer dataset, which represents a genuine medical diagnosis scenario with inherent positive-unlabeled structure rather than artificial conversion. The experimental results are presented in Tables 3 and 4 of our paper. Furthermore, we are committed to evaluating on additional real-world PU datasets and will include evaluation on the DIABETES dataset from TableShift as suggested by Reviewer k4vm.

---

> ### Comment · Area_Chair_xWnF · 2025-08-06
>
> Dear Reviewer,
>
> Thank you for submitting the Mandatory Acknowledgement. However, I noticed there is no visible engagement in the discussion thread. We kindly encourage you to engage with the authors, and clarify any outstanding issues.
>
> Thanks a lot.
>
> Best regards,
>
> AC

---

### Note · Authors · 2025-08-16

Dear Area Chair,

Thank you for facilitating this important discussion and providing us with the opportunity to address the reviewers' concerns.

We sincerely appreciate all the constructive feedback received throughout the review process. Based on the discussions, we would like to provide the following final clarifications:

1. **Regarding theoretical analysis**: We have successfully addressed the mathematical concerns in Lemma 3.1 and provided comprehensive theoretical explanations. The corrected bias analysis and the generalization bound in Theorem 3.3 directly explain the superior stability and performance observed in our experiments, as acknowledged by Reviewer YFBq who increased their score.

2. **Regarding experimental validation and methodology**: We have addressed concerns about experimental design, parameter sensitivity analysis, and the necessity of both weak and strong constraints in our dual-constraint mechanism, as appreciated by Reviewer cuct.

3. **Regarding real-world applicability**: We have successfully demonstrated our method's effectiveness on real-world scenarios, including the DIABETES dataset from TableShift and the Alzheimer dataset. Reviewer k4vm appreciated our comprehensive response and increased their score.

We are grateful that multiple reviewers have recognized the significant methodological contribution of our DC-PU risk estimator. The positive feedback and score improvements from multiple reviewers demonstrate the value and rigor of our research.

We thank you and the reviewers for the constructive feedback that has helped us present our contributions more clearly and comprehensively.

Sincerely,

The Authors

---

### Decision · Program_Chairs · 2025-09-17

**Decision:**

Accept (poster)

**Comment:**

This paper presents a well-motivated approach to addressing a long-standing issue in PU learning — imbalanced classification error rates between positive and negative classes.  The proposed method DC-PU introduces a dual-constraint mechanism that combines a dynamic lower bound and an equality constraint to maintain balanced empirical risks.

The reviewers generally agree that the paper is solid, with a clear motivation drawn from empirical observations, well-grounded theoretical analysis, and strong experimental results. A concern raised by multiple reviewers is that the theoretical section lacks sufficient intuitive explanation. The authors provided clarifications in the rebuttal. To improve the clarity and accessibility of the work, the revised version should include a more reader-friendly interpretation of the theoretical results within the main text. Overall, I find the paper to make a meaningful contribution to PU learning with both theoretical and practical impact, and I recommend acceptance.